# Requirement of DNMT1 to orchestrate epigenomic reprogramming for NPM-ALK–driven lymphomagenesis

Elisa Redl[1], Raheleh Sheibani-Tezerji[2], Crhistian de Jesus Cardona[3], Patricia Hamminger[4], Gerald Timelthaler[5], Melanie Rosalia Hassler[1,6], Maša Zrimšek[1], Sabine Lagger[7], Thomas Dillinger[1,2], Lorena Hofbauer[1,8], Kristina Draganić[1], Andreas Tiefenbacher[1,2], Michael Kothmayer[9], Charles H Dietz[10], Bernard H Ramsahoye[11], Lukas Kenner[1,7,12,13], Christoph Bock[10,14], Christian Seiser[9], Wilfried Ellmeier[4], Gabriele Schweikert[15,16], Gerda Egger[1,2]

**Malignant transformation depends on genetic and epigenetic events that result in a burst of deregulated gene expression and chromatin changes. To dissect the sequence of events in this process, we used a T-cell–specific lymphoma model based on the human oncogenic nucleophosmin-anaplastic lymphoma kinase (NPM-ALK) translocation. We find that transformation of T cells shifts thymic cell populations to an undifferentiated immunophenotype, which occurs only after a period of latency, accompanied by induction of the MYC-NOTCH1 axis and deregulation of key epigenetic enzymes. We discover aberrant DNA methylation patterns, overlapping with regulatory regions, plus a high degree of epigenetic heterogeneity between individual tumors. In addition, ALK-positive tumors show a loss of associated methylation patterns of neighboring CpG sites. Notably, deletion of the maintenance DNA methyltransferase DNMT1 completely abrogates lymphomagenesis in this model, despite oncogenic signaling through NPM-ALK, suggesting that faithful maintenance of tumor-specific methylation through DNMT1 is essential for sustained proliferation and tumorigenesis.**

## Introduction

Individual tumors and tumor types show a high level of heterogeneity regarding their genetic and epigenetic constitution and in the affected signaling pathways. Characteristically altered patterns of DNA methylation, however, are a universal hallmark of human cancer (1): In malignant cells, the genome is globally hypomethylated, whereas short CpG-dense regions, referred to as CpG islands (CGIs) generally show an increase in methylation (1, 2). CGIs are often found in gene promoter regions and hypermethylated CGIs have been associated with the silencing of tumor suppressor genes in diverse cancers. However, linking specific DNA methylation differences with extensive expression changes during tumorigenesis in a cause-and-effect relationship remains challenging because of the crosstalk of diverse epigenetic regulators. Furthermore, identifying the drivers that target the DNA methylation machinery is equally difficult. Studies in nucleophosmin-anaplastic lymphoma kinase (NPM-ALK) positive (ALK+) T-cell lymphoma, a subgroup of anaplastic large cell lymphoma (ALCL), have implicated the transcription factor STAT3 as a central player in epigenetic regulation (3, 4). STAT3 acts by directly and indirectly regulating the expression of the maintenance methyltransferase DNMT1 and by directing all three major methyltransferases (DNMT1, DNMT3A, and DNMT3B) to STAT3 binding sites within promoters of genes such as *SHP1* or *IL2RG*. The role of STAT3 as a mediator of DNA methylation of target promoters was supported by recent data, demonstrating a function of acetylated STAT3 for inducing the methylation of tumor suppressor genes in melanoma and breast cancer (5).

On the other hand, it has been recently suggested that disordered methylation patterns in tumors are resulting from stochastic processes and display intra-tumor heterogeneity, which could provide the basis for genetic and epigenetic tumor evolution (6, 7, 8). Interestingly, DNA methylation in tumors is frequently targeted to regions that are associated with H3K27me3 in embryonic stem cells (ESCs), resulting in an epigenetic switch from

[1]Department of Pathology, Medical University of Vienna, Vienna, Austria  [2]Ludwig Boltzmann Institute Applied Diagnostics (LBI AD), Vienna, Austria  [3]Eberhard Karls University of Tübingen, Faculty of Mathematics and Natural Sciences, Tübingen, Germany  [4]Division of Immunobiology, Institute of Immunology, Center for Pathophysiology, Infectiology and Immunology, Medical University of Vienna, Vienna, Austria  [5]Institute of Cancer Research, Medical University of Vienna, Vienna, Austria  [6]Department of Urology, Medical University of Vienna, Vienna, Austria  [7]Unit of Laboratory Animal Pathology, University of Veterinary Medicine Vienna, Vienna, Austria  [8]Research Institute of Molecular Pathology (IMP), Vienna Biocenter (VBC), Vienna, Austria  [9]Center for Anatomy and Cell Biology, Medical University of Vienna, Vienna, Austria  [10]CeMM Research Center for Molecular Medicine of the Austrian Academy of Sciences, Vienna, Austria  [11]Centre for Genetic and Experimental Medicine, Institute of Genomic and Molecular Medicine, University of Edinburgh, Edinburgh, UK  [12]Christian Doppler Laboratory for Applied Metabolomics (CDL-AM), Medical University of Vienna, Vienna, Austria  [13]Center for Biomarker Research in Medicine (CBmed), CoreLab 2, Medical University of Vienna, Vienna, Austria  [14]Department of Laboratory Medicine, Medical University of Vienna, Vienna, Austria  [15]Max Planck Institute for Intelligent Systems, Tübingen, Germany  [16]Division of Computational Biology, School of Life Sciences, University of Dundee, Dundee, UK

Correspondence: gerda.egger@meduniwien.ac.at

dynamic Polycomb repressed to more stable DNA methylation-based silencing (9, 10, 11, 12).

The impact of DNA hypomethylation has been widely studied by using *Dnmt1* hypomorphic alleles, which display reduced DNMT1 protein and activity levels (13, 14, 15, 16, 17, 18, 19, 20, 21, 22). Loss of DNA methylation and chromosomal instability seem to promote tumor initiation, whereas hypomethylation of tumor suppressor–associated CGIs exerts tumor-suppressive effects primarily during tumor progression. Such opposing effects of DNMT1 reduction were observed in different tumor models (19, 20). In addition, the de novo enzymes DNMT3A and B were shown to be involved in several hematological and solid cancers (23, 24, 25, 26, 27, 28, 29, 30). Thus, deregulation or mutation of DNMTs appears to be an essential event in tumorigenesis of various cancers.

In this study, we abrogated tumorigenesis in an NPM-ALK–driven T-cell lymphoma model by conditionally targeting the DNA methyltransferase *Dnmt1* by *Cd4*-Cre induced deletion. This allowed us to distinguish early NPM-ALK–driven events that occur independent of DNA methylation from later transcriptional changes that depend on the methylation machinery. We provide evidence for the cellular events associated with malignant transformation, which follow a period of latency and require the activation of MYC and NOTCH signaling pathways as well as pronounced epigenetic deregulation. Our findings provide further insight how oncogenes drive tumorigenesis by directing large scale transcriptomic and epigenomic alterations. This is an important prerequisite to identify potential therapeutic targets in ALK+ lymphoma and to gain a deeper understanding of large-scale epigenetic rearrangements that drive tumor transformation in general.

# Results

## Induction of ALK-dependent tumorigenic pathways following a period of latency

T-cell–specific expression of the human oncogenic fusion tyrosine kinase NPM-ALK under the control of the *Cd4* promoter/enhancer element in mice results in 100% transformation and tumor development at a median age of 18 wk (31). Notably, the time period before tumor onset and age of lethality is highly variable raising important questions about the nature and order of molecular events that need to occur during the latent phase to eventually trigger tumor initiation.

To better understand these steps, we first investigated the molecular state of NPM-ALK–induced tumors. We used genome-wide RNA sequencing (RNA-seq) to compare tumor cells in ALK transgenic mice with thymocytes isolated from age-matched wild-type mice. Differential gene expression analysis revealed that ALK tumor cells are characterized by a massive deregulation of gene expression with 2,727 genes significantly down- and 1,618 genes up-regulated as compared with control (Ctrl) cells (FDR-adjusted $P <$ 0.05, $\log_2$ fold change > 1) (Fig 1A and Table S1).

Given the large number of deregulated genes, it is extremely challenging to understand the role and importance of individual expression changes. To identify oncogenic pathways that are associated with ALK-dependent tumorigenesis, we first performed gene set enrichment analysis of significantly deregulated genes using oncogenic gene sets from the Molecular Signatures Database (MSigDB) (32, 33) (Fig 1B). Interestingly, the MYC pathway was among the top up-regulated pathways in ALK tumors compared with Ctrl thymocytes (Fig 1B and C), which we also confirmed using quantitative RT PCR (qRT-PCR) (Fig 1D). Besides *Myc*, we found the oncogene and Myc-regulator *Notch1* (34) as well as the MYC target genes and cell cycle regulators *Cdk4* and *Cdk6* to be significantly up-regulated in ALK tumors. This indicates that MYC signaling is involved in NPM-ALK–induced tumorigenesis, as previously observed in human ALK+ ALCL (35). Furthermore, we found cAMP signaling, which has a role for cell proliferation, differentiation and migration as well as the homeodomain-containing transcription factor HOXA9, which is implicated in hematopoietic stem cell expansion and acute myeloid leukemia, to be up-regulated in ALK tumors compared with Ctrl thymocytes (36, 37). In addition, TBK1, an AKT activator and suppressor of programmed cell death, was induced in ALK tumors (38). Among down-regulated gene sets, we found genes associated with the tumor suppressors *Atf2*, *Pten*, *Pkca*, *P53*, and *Rps14*, a negative regulator of *c-Myc*. Furthermore, genes associated with polycomb-repressive complex 1 (PRC1) were also down-regulated as well as mTORC1-regulated genes (39, 40, 41, 42, 43, 44). Together, these data suggest that ALK-induced transformation and lymphomagenesis involves the induction of additional oncogenic pathways and the repression of tumor suppressive genes.

## ALK+ tumor cells display an early double-negative (DN) immunophenotype

To get a better understanding of the order of events that lead from NPM-ALK expression to cancer, we next determined the immunophenotype of tumor cells. We used flow cytometry analysis (FACS) to characterize ALK-induced changes in cell composition in transgenic mice compared with control mice at different ages. To specifically analyze ALK+ cells, we combined an intracellular staining for NPM-ALK, with a classical surface staining protocol for common T-cell markers. We analyzed thymocytes isolated from thymi of 6- and 18-wk-old wild-type and ALK tumor-free mice, as well as tumor cells from ALK mice that had already undergone transformation (Fig 2A). We found that already in 6-wk-old transgenic mice almost 100% of T cells were ALK+, while showing no signs of altered thymus morphology. The expression levels of ALK showed a gradual increase from 6 to 18 wk and were highest in tumor cells compared with untransformed thymocytes (Fig 2A right panel).

Despite early expression of ALK, the distribution of T-cell subsets was normal in 18-wk-old ALK tumor-free mice as compared with Ctrls (Fig S1A). In ALK tumors of 18-wk-old mice, however, we observed the previously reported switch to CD4⁻CD8⁺ single-positive (SP) or CD4⁻CD8⁻ DN subsets (31). Furthermore, a similar fraction of Ctrls and ALK tumor-free thymocytes expressed the TCRβ, whereas the TCRβ⁺ fraction was absent in cells isolated from ALK tumors (Fig S1B). These two findings suggest that during the initial latent phase, T-cell

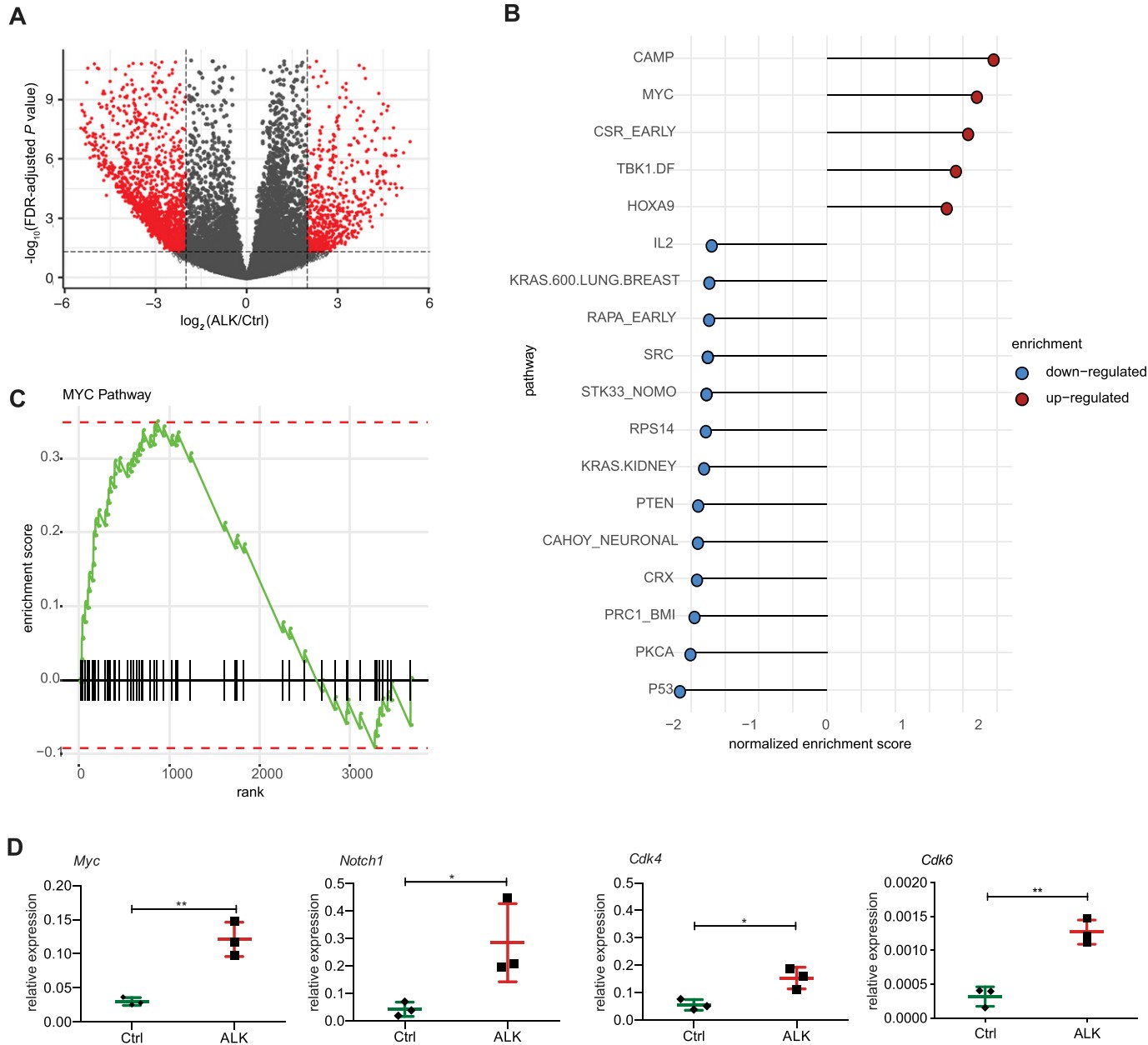

**Figure 1.  Deregulated gene expression in nucleophosmin-anaplastic lymphoma kinase (NPM-ALK) tumors.**
**(A)** Volcano plot displaying the differences in gene expression determined by RNA-seq between ALK tumor cells and wild-type (Ctrl) thymocytes, where red dots indicate significantly up- and down-regulated genes (permutation test followed by BH correction, FDR < 0.05, absolute $\log_2$(FC) higher than one) and grey dots show non-significantly altered genes (not meeting the criteria mentioned above) between these two groups. **(B)** Gene set enrichment analysis performed on the significantly up- and down-regulated genes filtered by FDR < 0.05 and absolute $\log_2$(FC) > 1 between ALK tumors and Ctrl thymocytes using oncogenic signature gene sets from MSigDB. Pathways associated with down-regulated genes are shown in blue and pathways associated with up-regulated genes are displayed in red ranked by normalized enrichment score. **(C)** Gene set enrichment analysis enrichment of MYC pathway-related genes among significantly deregulated genes filtered by FDR < 0.05 and absolute $\log_2$(FC) > 1 between ALK and Ctrl samples. The x-axis shows the differentially expressed genes belonging to the MYC pathway and the y-axis shows positive/negative enrichment scores for up-/down-regulated genes associated with the MYC pathway. **(D)** Analysis of MYC pathway related genes including *Myc*, *Notch1*, *Cdk4*, and *Cdk6* in Ctrl and ALK tumor samples using qRT-PCR. Analysis was performed in technical and biological triplicates. Data are represented as mean ± SD, *$P$ < 0.05, **$P$ < 0.01, using unpaired $t$ test. FC, fold change.

development progresses normally despite NPM-ALK expression and that the induction of T-cell transformation happens thereafter.

ALK+ T cells were also present in spleens from 18-wk-old ALK tumor-free mice as well as ALK tumor mice (Fig S2A). Before tumor

onset, these ALK+ cells were either CD4+ or CD8+ T cells (i.e., TCRβ+), whereas in tumor-bearing mice, additional CD4−CD8−TCRβ− tumor cells were detectable, suggesting that a fraction of ALK-transformed DN tumor cells was able to leave the thymus, or that the TCR

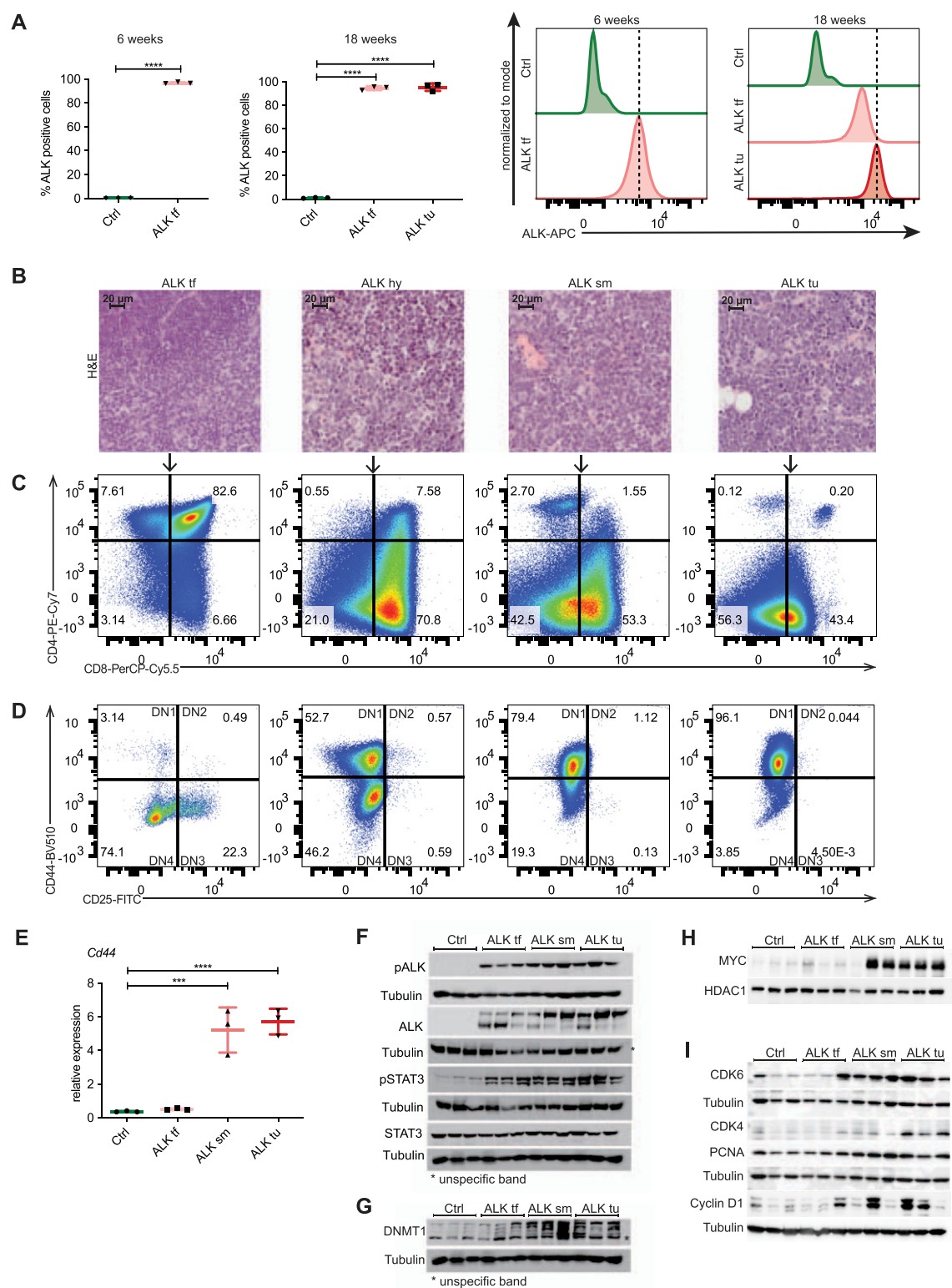

**Figure 2. Immunophenotype of ALK tumors.**
**(A)** Intracellular FACS analysis of ALK expression in thymocytes isolated from 6- to 18-wk-old wild-type (Ctrl), nucleophosmin-anaplastic lymphoma kinase (NPM-ALK) tumor-free (ALK tf) mice compared with ALK+ tumor cells (ALK tu). Quantification of the percentage of ALK+ cells in the three groups (left). Histograms (right) depict ALK expression levels compared with Ctrls. Dotted vertical lines indicate the peaks of ALK expression in ALK+ thymocytes of 6-wk-old ALK mice or ALK+ tumor cells at 18 wk of age. Data are represented as mean ± SD, ****P < 0.0001, one-way ANOVA, followed by unpaired *t* test, n = 3. **(B)** Hematoxylin and eosin (HE) stainings of representative thymi of 18-wk old NPM-ALK transgenic mice illustrating different stages of ALK tumors including a tumor-free thymus ALK tf; hyperplastic thymus, ALK hy; small tumor, ALK

expression is silenced after exit from the thymus as suggested previously (45) (Fig S2B).

## Tumor transformation and growth is accompanied by the occurrence of tumor cells with an immature T-cell profile

To further investigate the transformation process of thymocytes in NPM-ALK mice, we collected a series of thymi from NPM-ALK transgenic mice at 18 wk of age, which displayed different sizes and morphology correlating with different tumor stages as determined by histological analysis (Fig 2B). Immunophenotyping of these thymic samples revealed a gradual switch from CD4$^+$CD8$^+$ DP thymocytes to immature single positive thymocytes and DN subsets, correlating with tumor progression (Fig 2C). Further analysis of the DN population based on CD25 and CD44 expression revealed that in tumor-free thymi, most ALK+ cells were DN3 (CD25$^+$CD44$^-$) and DN4 (CD25$^-$CD44$^-$) stage thymocytes, which correlates with the developmental stages at which *Cd4* driven NPM-ALK expression was induced (Fig 2D). During tumorigenesis, there was a shift towards the DN1-like (CD25$^-$CD44$^+$) stage, which was also confirmed by qRT-PCR based on *Cd44* expression (Fig 2E). Together, these data suggest that ALK tumors are developing from a subpopulation of immature T cells or through reprogramming of DP T cells toward the DN stage based on the direct repression of the T-cell phenotype by NPM-ALK as previously suggested (46, 47).

## DNMT1 and STAT3 are activated during the latent phase

Next, we sought to identify events that occur in the latent phase before tumor initiation, which are potentially responsible for triggering the observed large-scale downstream reprogramming events. As already demonstrated by flow cytometry analysis, NPM-ALK and its active phosphorylated form pALK were detected by Western blot analysis independent of tumor presence, but with increasing protein levels correlating with tumor development (Figs 2F and S3). Global STAT3 levels were unaffected by ALK expression, however, its activated form pSTAT3 was significantly increased upon ALK induction independent of tumor presence, suggesting an early role in the latent phase before tumor onset. Previous work including our own has implicated epigenetic mechanisms in NPM-ALK–mediated lymphomagenesis in human cell lines and tumors in part through a direct effect of NPM-ALK and STAT3 signaling for the regulation and targeting of the major DNA methyltransferase DNMT1 (4, 46, 48, 49, 50, 51). Thus, we investigated DNMT1 protein levels before tumor onset and in different tumor stages from

tumor-free to end-term tumors in NPM-ALK transgenic mice in comparison to 18-wk-old wild types. Western blot analysis revealed a slight DNMT1 up-regulation already in tumor-free mice, which further increased during tumor progression (Fig 2G). Up-regulation of DNMT1 has been closely linked to the cell cycle and cell proliferation (52). To test whether the observed up-regulation of DNMT1 expression in ALK tumor samples is associated with deregulation of cell cycle–associated genes, we performed Western blot analysis using antibodies against c-MYC, CDK4/CDK6, cyclin D1, and the proliferation marker PCNA (Fig 2H and I). We detected a robust induction of c-MYC in nuclear extracts derived from small tumors and end-stage tumors compared with non-transformed thymi. In addition, cell cycle–related proteins, including CDK4/CDK6 and cyclin D1 were up-regulated upon tumor onset in thymi harboring small tumors, which was maintained in end-stage tumors for most of the samples tested. Together, these results suggest that ALK signaling is already active in pre-tumor stages and leads to an early activation of pSTAT3 and elevated expression of DNMT1 culminating in ALK-dependent transformation in NPM-ALK transgenic mice, which is accompanied by up-regulation of cell cycle genes and c-MYC induction.

## Deletion of *Dnmt1* abrogates NPM-ALK–dependent tumorigenesis

To investigate the functional role of DNMT1 for ALK-driven tumorigenesis in more detail, we intercrossed the *Cd4*-NPM-ALK transgenic mice (ALK) with mice carrying a T-cell–specific loss of *Dnmt1* (*Cd4*-Cre) (KO) (53). The resulting strain expressed the human *NPM-ALK* transgene but lacked a functional *Dnmt1* gene in T cells (ALKKO) (Fig 3A). It was shown previously that deletion of *Dnmt1* via the *Cd4* promoter in the double positive stage of T-cell development does not interfere with T-cell development (53), thus providing a suitable model to study the function of DNMT1 for ALK-dependent transformation of thymocytes. Strikingly, deletion of *Dnmt1* in this model completely abrogated ALK-driven lymphomagenesis as shown by Kaplan–Meier survival statistics (Fig 3B). The life span of ALKKO mice was identical to Ctrl and *Dnmt1* knockout mice, and no aberrant phenotype was detected in ALKKO thymi (Fig 3C). *Dnmt1* knockout in the context of ALK expression did not lead to changes in the relative percentages of DN, DP, and CD4 SP and CD8 SP thymocyte subsets (Fig S4). Pharmacologic inhibition of DNA methylation using 5-Aza-2′Deoxycytidine was also efficiently delaying tumor formation in the ALK model when administered from 8 to 30 wk of age (Fig S5). Effective deletion of *Dnmt1* in KO and ALKKO mice was confirmed at the protein level by

sm; end-stage tumor, ALK tu. **(B, C)** Representative FACS analysis of ALK+ cells isolated from 18-wk old tumor-free mice compared with different tumor stages (as in B) gated for CD4 and CD8 expression. **(B, D)** FACS analysis showing the expression of CD44 and CD25 to determine the different double negative (DN) stages of T-cell development (DN1-DN4) in ALK+ cells isolated from 18-wk-old tumor-free mice and different stages of ALK tumor developing mice (as in B). **(E)** qRT-PCR of *Cd44* expression in thymi of 18-wk old Ctrl and NPM-ALK tumor-free (ALK tf) transgenic mice as well as early developing tumors (ALK sm) and end-stage tumors (ALK tu) normalized to *Gapdh* expression. Analyses were performed in biological triplicates. Data are represented as mean ± SD, ***P < 0.001, ****P < 0.0001, using one-way ANOVA, followed by unpaired *t* test. **(F)** pALK, ALK, pSTAT3, and STAT3 protein levels in biological triplicates of thymi of 18-wk old Ctrl and NPM-ALK tumor-free mice as well as early and end-stage tumors were analyzed by Western blot analysis. Tubulin served as loading control. Asterisks indicate unspecific protein bands. **(F, G)** DNMT1 protein levels in biological triplicates of thymi of 18-wk-old Ctrl and tumor-free NPM-ALK transgenic mice as well as early and end-stage tumors (as in F) were analyzed by Western blot analysis. Tubulin served as loading control. Asterisks indicate unspecific protein bands. **(H)** Nuclear extracts were isolated from thymi of 18-wk-old Ctrl and NPM-ALK tumor-free mice and early and end-stage tumors in biological triplicates. Protein levels of MYC were detected by Western blot analysis. The nuclear protein HDAC1 served as loading control. **(I)** Protein levels of the cell cycle associated genes CDK4, CDK6, PCNA, and cyclin D1 in biological triplicates of thymi of 18-wk-old Ctrl and NPM-ALK tumor-free mice as well as early and end-stage tumors were examined using Western blot analysis. Tubulin served as loading control.

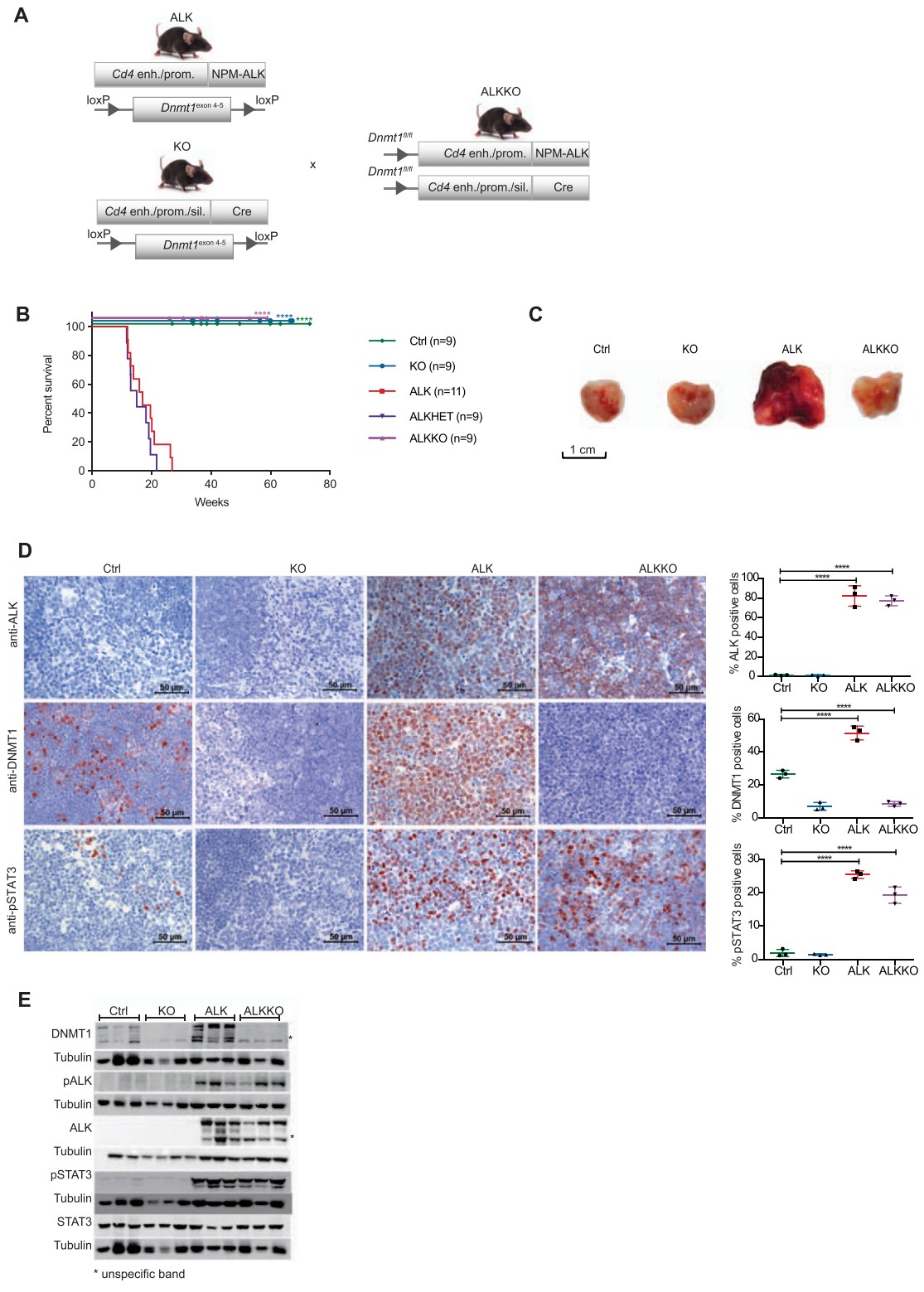

**Figure 3.   Deletion of *Dnmt1* abrogates lymphomagenesis in nucleophosmin-anaplastic lymphoma kinase (NPM-ALK) transgenic mice.**
**(A)** Generation of mice with T cell-specific *Cd4*-NPM-ALK expression (ALK) and T cell-specific deletion of *Dnmt1* (KO) or both (ALKKO). *Cd4* enh./prom., *Cd4* enhancer and promoter. *Cd4* enh./prom./sil., *Cd4* enhancer, promoter and silencer. **(B)** Kaplan-Meier survival statistics depicting overall survival of *Cd4*-NPM-ALK (ALK), *Cd4*-NPM-ALK *Cd4*-Cre *Dnmt1*^flox/+ (ALKHET), *Cd4*-NPM-ALK *Cd4*-Cre *Dnmt1*^flox/flox (ALKKO), and *Dnmt1*^loxP/loxP control mice (Ctrl). ****P < 0.0001, Log-rank (Mantel–Cox) test, pairwise comparison to ALK. **(C)** Morphology of 18-wk-old Ctrl, KO and ALKKO thymi in comparison to ALK tumors. Pictures were taken immediately after organ collection.
**(D)** Protein expression of NPM-ALK, DNMT1, and pSTAT3 was analyzed by immunohistochemistry staining in Ctrl, KO, ALKKO thymi, and in ALK tumors. Pictures are

immunohistochemistry (IHC) and Western blot analysis (Fig 3D and E). Notably, abrogation of DNMT1 did not interfere with ALK levels or activity in ALKKO transgenic mice as indicated by similar pALK levels in ALK tumor cells and ALKKO thymocytes nor did it change pSTAT3 levels (Fig 3E). Thus, our data suggest that depletion of DNMT1 interrupts the chain of events that leads from ALK activation to T-cell transformation.

## *Dnmt1* deletion reduces proliferation of ALKKO cells

We next studied the proliferative capacity of ALKKO thymocytes compared with the other genotypes by assessing Ki67 expression. As expected, ALK tumor cells showed significantly higher prolif- eration rates than thymocytes of Ctrl, KO, and ALKKO mice (Fig 4A). A trend towards lower proliferation than in Ctrl and KO was de- tectable in ALKKO thymi. To specifically analyze ALK-expressing cells, we next performed co-staining of ALK and Ki67 in ALK and ALKKO tissues using immunofluorescence (Fig 4B). Quantification of ALK and Ki67 double-positive cells revealed that ALK-expressing cells showed a significant reduction of Ki67 positivity in ALKKO thymi compared with ALK tumors, with 85.7% double positive in ALK versus 58.7% in ALKKO samples. In particular, ALKKO cells with high ALK expression levels showed low expression of Ki67, whereas cells with high Ki67 levels showed very low expression of ALK, indicating that ALK cells cannot induce or maintain stable proliferation upon *Dnmt1* deletion.

## *Dnmt1* knockout inhibits ALK-dependent transcription programs

To investigate the molecular mechanisms associated with *Dnmt1* deletion in ALK transgenic mice, we performed RNA-seq analyses of thymocytes isolated from Ctrl, KO, and ALKKO mice and compared it with our tumor RNA-Seq data (Fig 5). Sample distance and principal component analysis revealed a clear separation of ALK tumors from all other genotypes, which were clustering together (Figs 5A and S6A). Within the groups, ALK tumors showed the largest hetero- geneity. Along these lines, unsupervised clustering of the 5% most variably expressed genes revealed a clear separation of tumor samples from all other genotypes, which showed highly similar expression patterns (Fig 5B). Differential gene expression analysis showed that ALKKO displayed highly similar gene expression patterns compared with Ctrl cells: We only found 8 genes to be significantly down- and 99 genes up-regulated (Figs 5C and S6B and Table S2). When compared with ALK tumor samples, ALKKO was similar to KO and Ctrl samples and showed a high degree of de- regulation with 2,819 genes up- and 1,355 genes down-regulated (Fig S6B).

We were particularly interested in a small group of genes, which were consistently deregulated in both ALK+ cell types relative to Ctrl and *Dnmt1* KO cells, as identified by pairwise comparisons of gene expression differences because they might constitute direct targets

of NPM-ALK upstream of DNMT1 (Fig S6C). Apart from ALK, we found seven genes (*Tha1*, *Trim66*, *Gzma*, *Socs3*, *Gm5611*, *5830468F06Rik*, and *Gm17910*), which were up-regulated both in ALK and ALKKO cells compared with Ctrl. These include the suppressor of cytokine signaling 3 (*Socs3*), which is a regulator of the JAK/STAT signaling pathway and was also found up-regulated in human ALK+ ALCL cell lines (54), tripartite motif containing 66 (*Trim66*), which is part of the rat sarcoma (RAS) pathway that regulates DNMT1 expression and is known to promote proliferation (55, 56, 57) and granzyme A (*Gzma*) a canonical cytotoxic gene that is involved in cancer initiation and progression (58) (Fig 5D). In addition, strong up-regulation of *c-Myc*, *Cdk4/6*, and *Cyclin D1* was detectable in ALK tumors at the RNA level, which was in concordance with elevated protein levels of these cell cycle regulators (Fig S6D). These data suggest that expression of ALK initially affects a small number of regulatory genes including *Socs3*, *Trim66*, and *Gzma*, whereas a downstream substantial rewiring of the whole transcriptional program eventually accompanies ma- lignant transformation.

The large heterogeneity observed in biological replicates of ALK tumors as well as the results from FACS analyses suggested a change in cell composition in ALK tumors compared with the other genotypes. Thus, we established a deconvolution strategy, which allowed us to infer different thymic cell populations in our data from previously published thymic single-cell RNA-seq (scRNA-seq) datasets (59). Using this strategy, along with the relative expression of marker genes in our bulk data, we were able to show that cell proportions of our Ctrl samples were comparable with thymi of 4- to 24-wk-old mice used in the study by Park et al (Fig 6A). Likewise, KO and ALKKO samples showed similar cell type proportions, suggesting little changes in cell composition upon *Dnmt1* deletion in normal as well as ALK-positive thymocytes (Fig 6B). The one exception was the KO2 sample, which appeared to be an outlier in this analysis. Strikingly, we observed an increase in quiescent DN cells (DN(Q)) and a slight decrease in SP CD4 and CD8 cells based on thymocyte-specific marker gene expression in ALK tumor samples (Fig 6B). Interestingly, when we included scRNA-seq datasets and marker genes from early murine thymus develop- ment in the deconvolution analysis, we observed a higher con- cordance of marker gene expression patterns of ALK tumors with early embryonic thymocytes compared with mature thymocytes (Fig 6C).

A caveat of our analysis in the case of the tumor sample was the use of T-cell–specific scRNA-seq data for the deconvolution of bulk tumor samples, which might contain additional cell types (e.g., tumor cells) not present in the single cell data. To get a better understanding of the genes expressed in the tumor cell population that are not accounted for in the deconvolution, we applied the following strategy. The deconvolution of the ALK data was based on 175 specific marker genes determined on the single cell data (see the Materials and Methods section). We then simulated genome- wide expression patterns of a cell mixture similar to the one

representatives of biological triplicates. Graphs below the images depict quantification of stainings using Definiens Tissue Studio 4.2 software. Data are represented as mean ± SD, ****P < 0.0001, using one-way ANOVA, followed by unpaired *t* test. **(E)** DNMT1, pALK, ALK, pSTAT3, and STAT3 protein levels in thymi of 18-wk-old Ctrl, KO, and ALKKO mice as well as ALK tumors were analyzed by Western blot analysis. Tubulin served as loading control. Analysis was performed in biological triplicates. Asterisks indicate unspecific band.

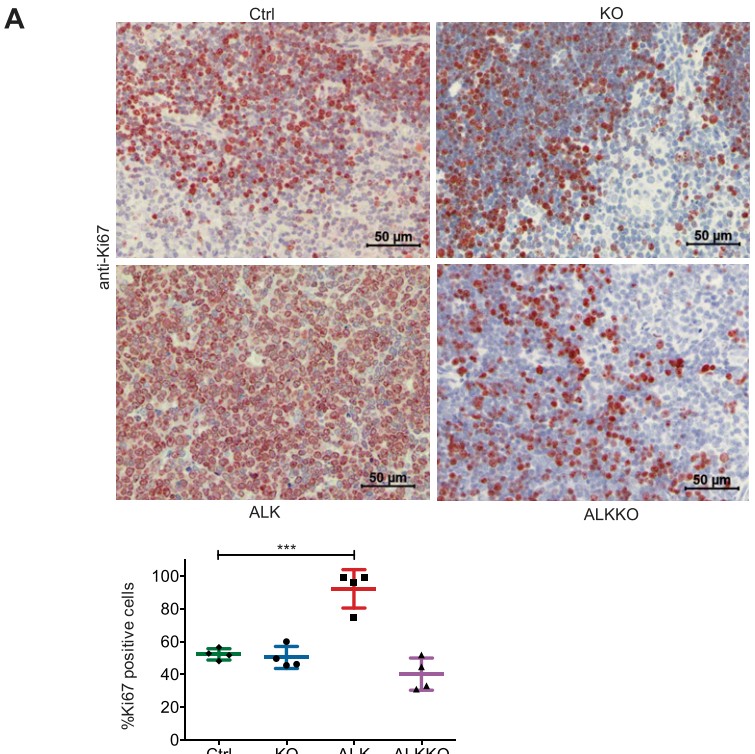

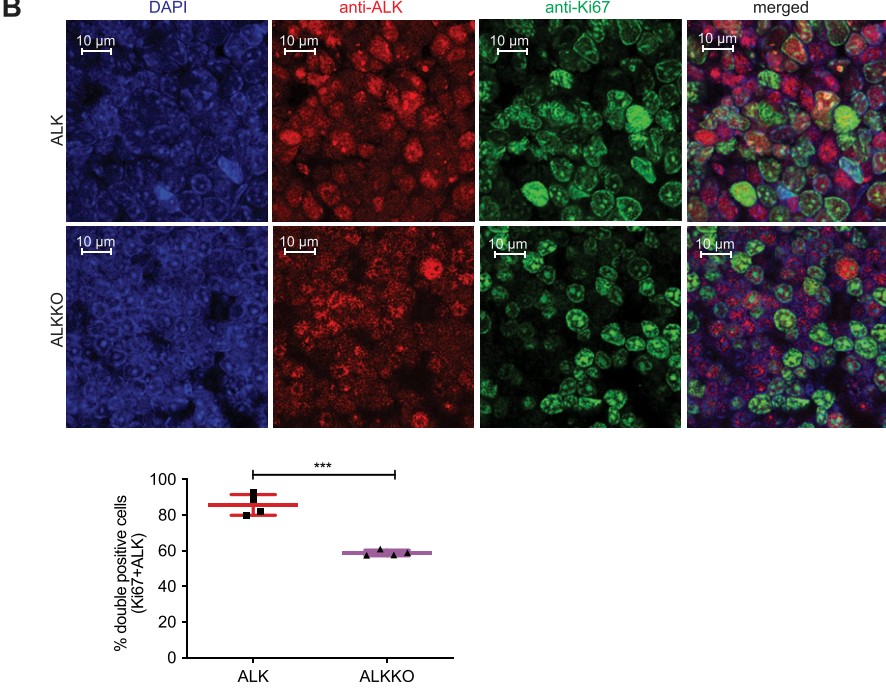

**Figure 4. Deletion of *Dnmt1* results in reduced proliferation of ALK+ cells.**
**(A)** Cell proliferation analysis by immunohistochemistry staining of Ctrl, KO, ALKKO thymi and ALK tumor tissues using Ki67 antibody. The graph shows quantification of Ki67 positive cells using Definiens Tissue Studio 4.2 software. Non-proliferative areas in the thymus were excluded from analysis. Data are shown as mean ± SD, ***P < 0.001, pair-wise comparison to control using unpaired *t* test, n = 4. **(B)** Double immunofluorescence staining of ALK tumors and ALKKO thymi. Tissues were stained with antibodies against ALK (red) and Ki67 (green) and counterstained with DAPI (blue). Pictures were acquired with identical pixel density, image resolution, and exposure time. The graph shows quantification of immunofluorescence staining by counting Ki67/ALK double-positive relative to total number of cells (DAPI positive) of two equally sized areas per tumor/thymus from four biological replicates, respectively. Cell counting was performed by two individuals and slides were blinded for counting. Data are shown as mean ± SD, ***P < 0.001, using unpaired *t* test.

expected in the ALK sample if it was only composed of the seven accounted cell types (simALK). We assumed that if there are extra cell types in the ALK tumor replicas these additional cell types should be characterized through differentially expressed genes between the simulations and the real replicas. Thus, we chose the top 10% most highly expressed genes in the original bulk data plus the 175 marker genes and performed a differential expression analysis between the bulk samples and the corresponding replicas in the simulation. Notably, the 175 maker gens showed the largest expression changes in the simulated versus real data, suggesting

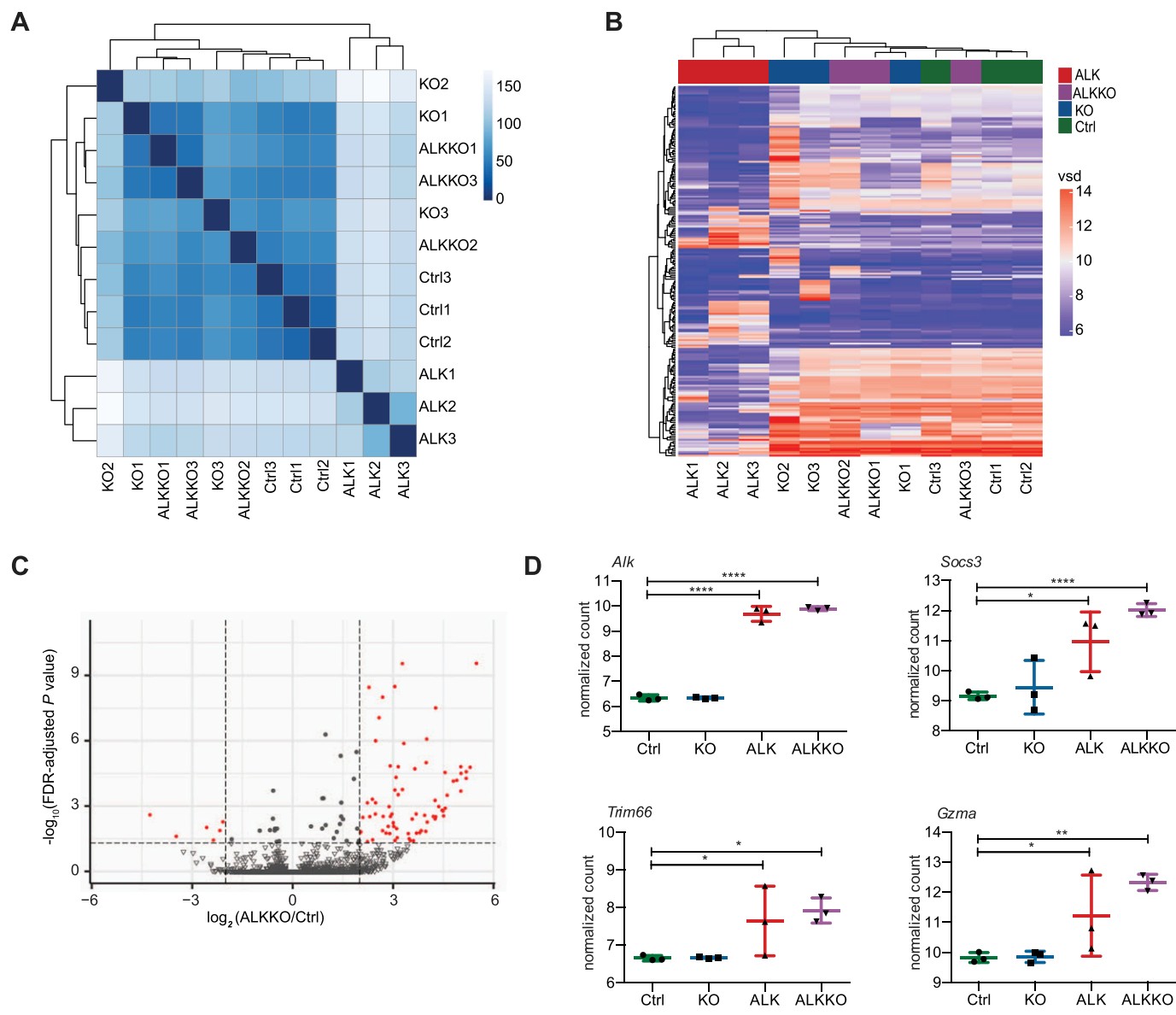

**Figure 5. High similarity in gene expression between ALKKO and Ctrl thymocytes.**
**(A)** Hierarchical clustering heat map illustrating sample to sample Euclidian distances based on variance stabilizing transformations of RNA-seq gene expression values of all genes of individual Ctrl, KO, ALKKO thymi, and ALK tumor samples. **(B)** Heat map showing unsupervised clustering of the top 5% most variable genes among all samples in Ctrl, KO, ALK, and ALKKO using variance stabilizing data. **(C)** Volcano plot displaying the significant differences in gene expression between ALKKO compared with Ctrl thymocytes, where red dots indicate significant differentially expressed genes (FDR < 0.05, absolute $\log_2$(FC) > 1), and grey scale dots show nonsignificant differentially expressed genes, between these two groups. **(D)** Gene expression levels of *Socs3*, *Trim66*, and *Gzma* based on normalized counts from RNA-seq analysis of ALK tumor and ALKKO thymus samples compared with Ctrl and KO thymi. Data are represented as mean ± SD, *P < 0.05, **P < 0.01, ****P < 0.0001, using ordinary one-way ANOVA followed by multiple comparison using Fisher's least significant difference (LSD) test.

the presence of additional cell types (Fig 6D). For the remaining genes, we found genes associated with T-cell activation, such as *Cd6* and *Cd69*, and interferon response, such as *Isg15*, *Slfn2*, or *Ifit1bl1* to have lower expression than expected from the simulated T-cell mixture, indicating substantial expression changes even in the considered cell types (Table S3). Furthermore, genes that showed higher expression patterns in ALK tumors compared with the simulated mixture were associated with cancer development, progression and aggressiveness, such as *Tbx19*, *Nwd1*, *Sytl2*, *Amigo2*, and *Dpp4*. Genes that were associated with Notch signaling (*Nrarp*

and *Dtx1*) were also strongly induced in ALK tumors compared with the simulated ALK sample (Table S4). Together, these data support our previous findings suggesting an enrichment of DN cells, downregulation of T-cell–specific factors and a strong induction of oncogenic pathways in ALK-driven tumors.

### Changes in DNA methylation after *Dnmt1* deletion

To assess the effect of DNMT1 depletion on global DNA methylation in the different genotypes, we analyzed global and site-specific CpG

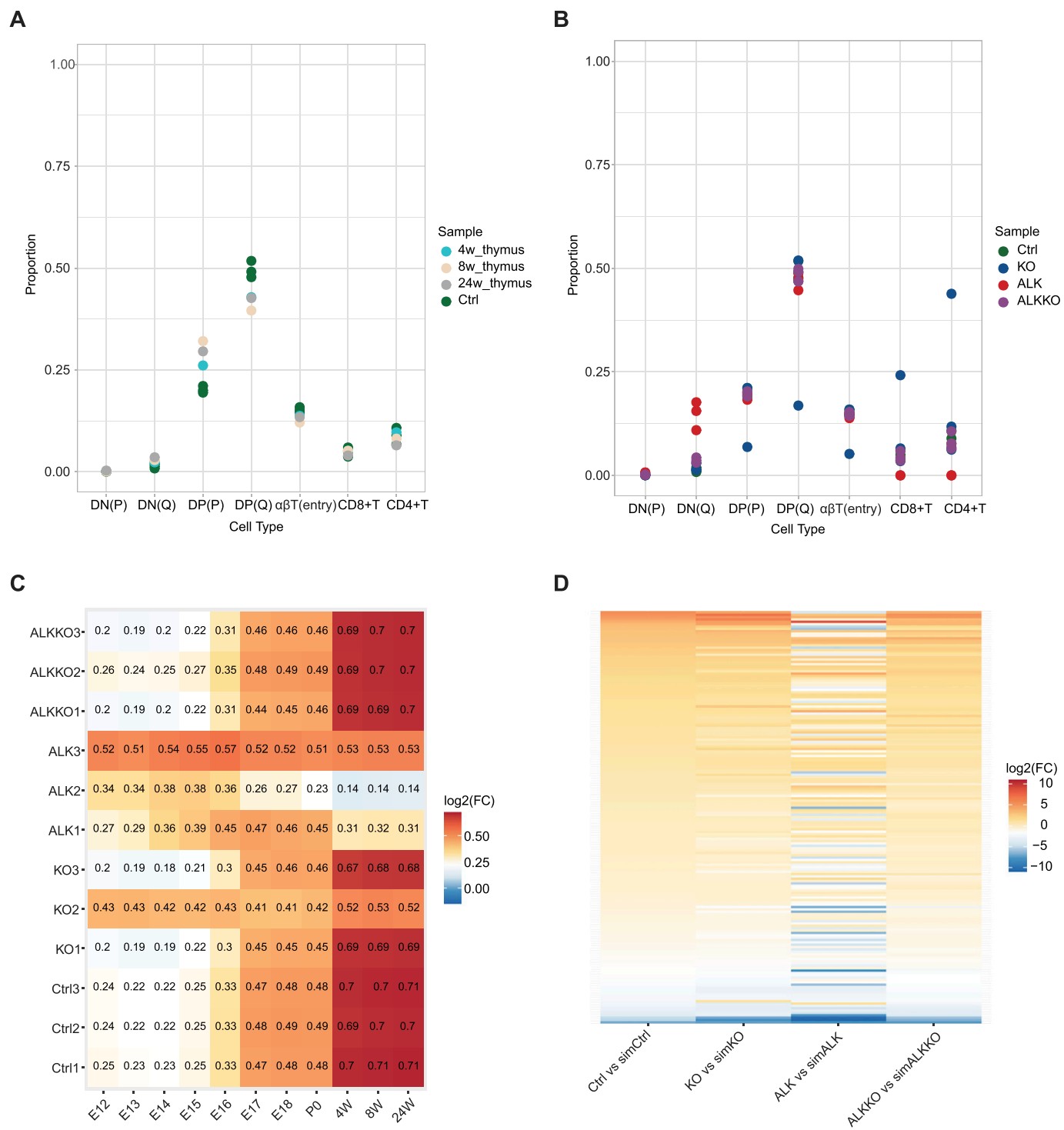

**Figure 6. Deconvolution of RNA-seq data.**
**(A)** Proportions of cell types in the sc-RNA-seq data in postnatal thymus harvested at 4, 8, and 24 wk of age compared with the proportions calculated for the deconvolution using the bulk data (Ctrl sample). DN, double-negative T cells; DP, double-positive T cells; P, proliferating; Q, quiescent. **(B)** Estimation of the cell type proportions in each sample in the bulk data. KO2 appears to be an outlier in this analysis. **(C)** Correlation between each replica in the bulk data and the single cell data of different time points using the marker genes selected in the deconvolution. E, embryonic day; P0, birth; W, weeks after birth. **(D)** Log₂(FC) values of the real versus simulated samples for the 175 marker genes selected in the deconvolution.

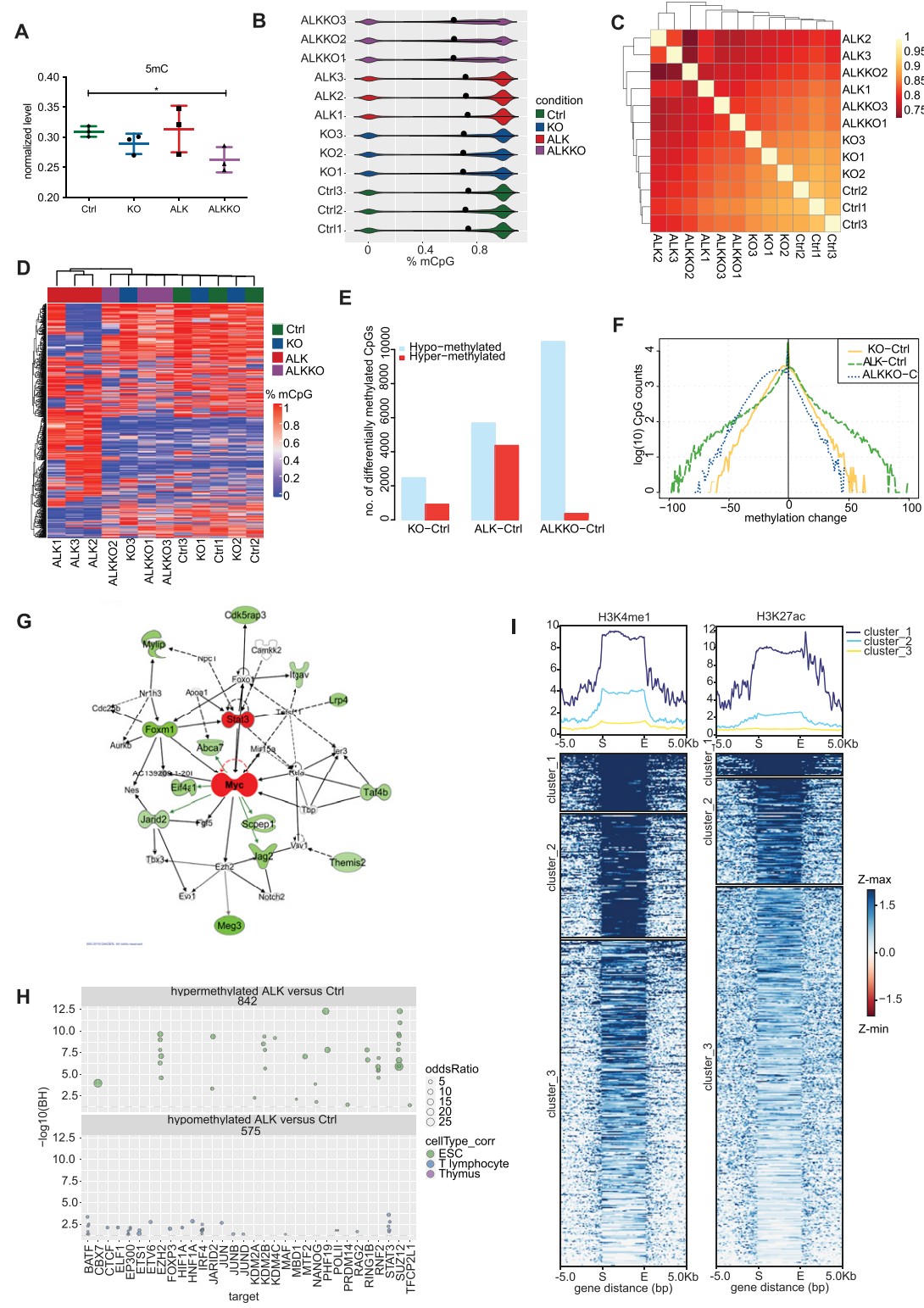

**Figure 7.   DNA methylation changes in ALK tumors and *Dnmt1* knockout thymi.**
**(A)** Quantification of global DNA methylation levels by dot blot analysis using 5mC immunodetection. 5mC signal intensities were normalized to total DNA input based on methylene blue staining. Data are represented as mean ± SD, *P < 0.05, using one-way ANOVA, followed by unpaired t test, n = 3. **(B)** Violin plots indicating the bimodal distribution of methylation levels determined by reduced representation bisulfite sequencing in three biological replicates of Ctrl, KO and ALKKO thymi as well as ALK tumors. Shown are percent methylation per CpG (%mCpG). Black dots indicate the median percentage of mCpG in each sample. **(C)** Correlation heat map between individual Ctrl, KO, ALKKO and ALK samples based on DNA methylation levels of single CpGs based on correlation coefficients between samples. **(D)** DNA methylation heat

methylation, using dot blot analysis measuring 5-methyl Cytosine (5mC) levels and genome-wide methylation patterns using reduced representation bisulfite sequencing (RRBS), respectively (Figs 7A and B and S7A). As shown previously (53), KO samples revealed a slight but not significant decrease in global DNA methylation compared with Ctrl samples. In the double mutant, ALKKO, we observed a significant decrease in global DNA methylation levels as compared with Ctrl samples. The loss of DNA methylation in KO and ALKKO samples was accompanied by strong induction of *IAP* retrotransposon transcription, which was especially high in ALKKO, in line with higher loss of global methylation in those samples (Fig S7B).

CpG methylation showed the highest correlation between Ctrl and KO samples, whereas both ALK and ALKKO samples showed lower correlations in relation to other genotypes but also amongst the replicates (Fig 7C). Similarly, unsupervised hierarchical clustering of the top 1% most variable CpGs indicated a separation of ALK tumor samples compared with Ctrl, KO, and ALKKO thymi and a closer proximity between Ctrl and KO compared with ALKKO samples, suggesting that the largest changes in methylation occur in the tumor samples compared with the other genotypes (Fig 7D). Compared with Ctrl, ALK samples showed both hyper- and hypomethylation, whereas the KO and ALKKO samples were enriched for hypomethylated sites (Fig 7E). When comparing the change in methylation per site, we observed that in ALKKO compared with Ctrl, most sites lost less than half of their methylation, whereas in ALK versus Ctrl we detected CpGs with particularly large changes reflecting stable tumor-specific changes in DNA methylation patterns (Fig 7F). Annotation of differentially methylated CpGs revealed that ALK tumor samples mainly gained DNA methylation in promoter regions, CGIs and shores, whereas intergenic and inter-CGI sites preferentially lost methylation (Fig S7C and D). The large number of hypomethylated CpGs between Ctrl and ALKKO was mainly associated with intergenic and inter-CpG island regions and to a lesser extent with upstream promoter regions, exons, shores and shelves (Fig S7E and F).

To identify potential tumor-relevant pathways associated with differentially methylated regions (DMRs), we performed ingenuity pathways analysis (IPA, www.qiagen.com/ingenuity), interrogating promoters that showed tumor-specific hypomethylation. Interestingly, those analyses identified c-MYC as a significant upstream regulator and several genes connected to the cellular c-MYC network were hypomethylated in ALK tumor samples (Fig 7G). Among these genes, *Cdk5rap3*, *Lrp4*, *Eif4a1*, and *Taf4b* were also significantly up-regulated in tumors compared with Ctrls based on RNA-seq data (Table S1). Interestingly, all four genes have been implicated in proliferation control and tumorigenesis before (60,

61, 62, 63), highlighting their potential relevance for ALK driven lymphomagenesis.

To get further information about the genomic regions that are affected by aberrant methylation in ALK tumor cells we performed genomic region enrichment analysis on differentially methylated promoter-associated CpGs tiled into 200-bp genomic regions using locus overlap analysis (LOLA) (64). We used published ChIP-Seq datasets integrated in the LOLA software tool including murine ESC, T lymphocytes, and thymus for comparison with our DNA methylation profile of ALK tumor cells. Hypermethylated promoter regions in ALK tumor cells showed strong enrichment of binding sites for proteins associated with chromatin remodeling in ESC, in particular PRC associated proteins including CBX7, EZH2, MTF2, PHF19, RING1B, RNF2, and SUZ12. In addition, we detected a significant overlap of hypermethylated regions with ESC-specific transcriptional regulators of pluripotency, including NANOG, PRDM14, and TFCP2L1 (Fig 7H). Epigenetic switching from PRC marks to DNA methylation is a well-described phenomenon in human tumor cells reducing epigenetic plasticity of tumor cells (9, 65). For hypomethylated regions, we identified several transcription factors with T-cell–specific functions, some of which have been implicated in human ALCL before. Those included AP-1 family members BATF, JUN, JUNB, JUND, or IRF4 and STAT3, all of which have been associated with NPM-ALK signaling and tumorigenesis (66, 67, 68, 69, 70). Interestingly, motif enrichment using the Analysis of Motif Enrichment (71) integrated in the MEME Suite tool (72) revealed significant enrichment of several transcription factor motifs in the DMRs previously identified in the LOLA analysis including STAT3, ELF1, ETS1, or FOXP3 (Table S5).

Enhancers play an important role in regulating gene expression during development and it has been shown that enhancer methylation can be drastically altered in cancer, which can be associated with altered expression profiles of cancer genes (73). Therefore, we compared DMRs, defined over 1-kb tiling windows, between ALK tumor cells and Ctrl thymocytes to ENCODE ChIP-seq datasets for active enhancers including monomethylation of histone H3 lysine K4 (H3K4me1) and acetylation of histone H3 lysine 27 (H3K27ac) in murine thymi (74, 75). Among 470 DMRs tested, we detected a specific overlap of 118 DMRs with H3K27ac and 167 DMRs with H3K4me1, respectively (Fig 7I). Of these, 95 DMRs overlapped with both H3K4me1 and H3K27ac, indicative of active enhancers in thymocytes. Interestingly, several genes nearby differentially methylated enhancers showed significantly deregulated gene expression (Table S6). Among those genes, we found *Runx1*, a critical transcription factor for early T-cell development of thymic precursors and T-cell maturation (76, 77), or *Socs3*, which was also

---

map using unsupervised clustering of the top 1% most variable CpGs in Ctrl, KO, ALK and ALKKO samples. The color code represents mean methylation levels of individual CpG sites. **(E)** Significantly hyper- (red) and hypo- (blue) methylated CpGs resulting from comparison of KO, ALKKO, and ALK versus Ctrl samples identified by methylKit analysis (*P* < 0.01; b-value difference >25%). **(F)** Methylation changes of individual CpGs relative to Ctrl for all genotypes (as in E) shown as density plot. Numbers on the y-axis are log(10) CpG counts. **(G)** Network analysis based on Ingenuity Pathway Analysis of significantly hypomethylated promoter regions in ALK versus Ctrl samples. Significantly hypomethylated genes are depicted in green, upstream regulators are depicted in red. **(H)** Locus overlap analysis (LOLA) region set enrichment analysis for differentially methylated CpGs (binned into 1-kilobase tiling regions). The plot shows region sets from embryonic stem cells (green), T lymphocytes (blue) and thymus (purple) with *P* < 0.05. **(I)** Differentially methylated regions (DMRs) between ALK versus Ctrl samples were used to map histone modifications (H3K4me1 and H3K27ac) of ENCODE regions defined by ChIP-seq data of thymus samples using *k*-means clustering (*k* = 3). The heat map indicate overlap of individual DMRs expanded to 5 kb on both sites with H3K4me1 (left) and H3K27ac (right) in three distinct clusters, based on different signal intensity. Gene distance indicates predicted enhancers mapped within at a 5-kb window indicating start (S) and end (E) of the DMR region. Z-min/max shows the intensity of the H3K4me1 and H3K27ac ChIP-seq signals.

**A**

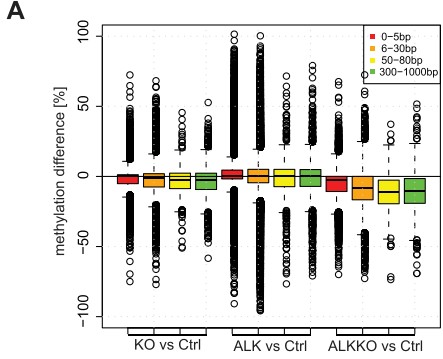

**B**

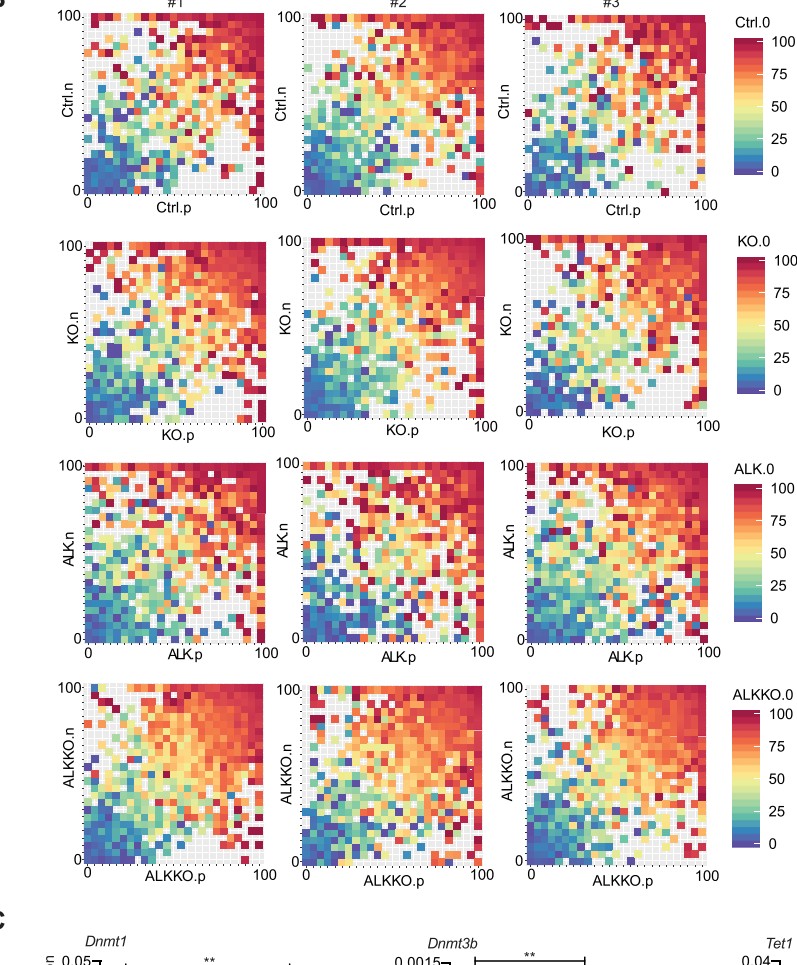

**Figure 8. Analysis of associated DNA methylation.**
**(A)** Methylation difference of neighboring CpGs relative to Ctrls in KO, ALK, and ALKKO samples as analyzed based on the distance between CpGs. **(B)** CpG triplet analysis of neighboring CpGs within a maximal distance of 20 bp. Each square represents three neighboring CpGs. The color of the squares indicates the methylation level of the middle CpG from unmethylated (blue) to fully methylated (red). The x and y axes represent the methylation levels of the two neighboring CpGs annotated as (n) and (p). Squares close to the diagonal indicate highly correlated CpG triplets, whereas dispersed squares represent non-correlated triplets. **(C)** qRT-PCR of *Dnmt1*, *Dnmt3b*, and *Tet1* expression in Ctrl, KO and ALKKO thymocytes as well as ALK tumor cells normalized to *Gapdh* expression. Analyses were performed in biological triplicates. Data are represented as mean ± SD, *$P <$ 0.05, **$P <$ 0.01, pairwise comparison to the Ctrl unpaired $t$ test.

**C**

implicated in human ALCL (78, 79). Down-regulated genes included the phosphatase and tumor suppressor *Ptpn13* (80) and the TGF-β ligand *Bmp3*, which might result in altered TGF-β signaling as observed in human ALK-positive cancers (81).

Together, these data suggest that NPM-ALK–driven transformation is accompanied by CGI hypermethylation and hypomethylation of intergenic regions, which is reminiscent of epigenetic reprogramming events in human tumors and affects major oncogenic signaling pathways.

Furthermore, we found associations of tumor related hypermethylation with PRC1- and PRC2-regulated regions, as well as a significant overlap of hypomethylated regions with T-cell–specific and tumor-relevant transcription factor–binding regions and motifs and thymocyte specific enhancers.

### Loss of associated methylation patterns in tumors

Recent literature has challenged the standard model of DNA methylation inheritance, in which CpGs are presumed as independent but has rather suggested a collaborative model that takes methylation levels of neighboring CpGs into account (82, 83). According to this model, which also considers the distances of neighboring sites, methylated CpGs would enforce methylation of adjoining CpGs, whereas non-methylated CpGs would induce rather unmethylated states depending on the presence of DNMTs and TET enzymes. We therefore analyzed whether the distance between two neighboring CpGs had an influence on methylation changes of those CpGs in ALK tumor or knockout cells (ALK and ALKKO) relative to Ctrls (Fig 8A). ALK tumor cells showed stronger hypermethylation in nearby CpGs (0–5 bp distance), most likely reflecting tumor specific CGI hypermethylation. More distant sites showed equal gains and losses of methylation in the ALK samples. Furthermore, we found that in the KO samples as well as in the double mutant ALKKO, highest methylation loss was observed at CpGs more distant from neighboring CpGs, suggesting that DNMT1 is needed for methylation maintenance of distant CpGs.

We further examined closely associated CpG triplets, defined as CpGs with neighboring CpGs within less than 20 bp distance to both sites, in three biological replicates of each genotype (Fig 8B). We found that in Ctrl samples methylation levels between neighboring sites are highly correlated, in a way that a highly methylated CpG is flanked by highly methylated CpGs, whereas unmethylated CpGs are flanked by unmethylated CpGs. Thus, in a correlation plot, most CpGs are found close to the diagonal in Ctrl samples (Fig 8B). *Dnmt1* knockout samples showed a general trend towards lower methylation but retained the correlation among the CpG triplets, as illustrated by the vicinity of CpG triplets to the diagonal in KO and ALKKO samples, suggesting that DNMT1 is not required to warrant cooperative methylation. Interestingly, cooperative DNA methylation seems to be disturbed in ALK tumor cells, where we detected a loss of correlation and observed a higher methylation difference of neighboring CpGs, indicating a loss of collaboration between those sites.

Together, these data suggest that heterogeneous DNA methylation patterns in ALK+ tumors are characterized by low correlation of DNA methylation between nearby CpGs, which might be associated with the deregulation of methylation controlling enzymes including *Dnmt1*, *Dnmt3b*, and *Tet1* (Fig 8C) that we observed based on qRT-PCR analyses. Interestingly, the correlation between neighboring CpGs is maintained upon DNMT1 depletion, suggesting that de novo methyltransferases such as DNMT3a/b can compensate and control associated DNA methylation at sites in close vicinity.

## Discussion

The ALK protein has been identified as a driver oncogene in diverse cancers, based on its overexpression after genetic translocation,

amplification, or mutation (84, 85, 86). Overexpression of the human NPM-ALK oncogene in mouse T cells results in lymphomagenesis with 100% penetrance. In this model, the relatively long period of latency despite NPM-ALK expression and activation of downstream targets such as STAT3 and DNMT1 suggests that a secondary event is necessary for transformation and tumorigenesis. Accordingly, the NPM-ALK translocation and its transcripts can be detected in peripheral blood cells of human healthy donors implying that a single genetic event is not sufficient to induce transformation (87). Both our transcriptomic and epigenomic data point towards MYC as the potential secondary driver in this model. MYC was previously implicated in different ALK dependent malignancies including non-small cell lung cancer neuroblastoma and ALK+ ALCL (35, 69, 88, 89, 90, 91, 92). Furthermore, MYC was shown to directly regulate expression of *Cdk4* and *Cdk6*, affecting cell cycle progression at multiple points (93). We observed a gradual up-regulation of those genes on mRNA and protein level from tumor initiation to end-stage tumors. Correspondingly, we observed strong deregulation of *Notch1* in the transgenic mouse model. The NOTCH-MYC axis plays a major role in the development of T-cell acute lymphoblastic leukemia (T-ALL), resulting from the transformation of immature T-cell progenitors (94). This suggests that similar processes are driving ALK-dependent transformation, resulting in hyperproliferation and transformation of thymocytes. Notably, deletion of *Dnmt1* in an MYC-driven T-cell lymphoma model delayed lymphomagenesis and resulted in reduced proliferation of tumor cells (95).

During lymphomagenesis, we observed a gradual decrease in DP thymocytes subsets and a corresponding increase in DN and immature CD8 SP (i.e., TCR$\beta^-$) cells. Similarly, human ALK+ ALCL display a progenitor cell signature as indicated by epigenomic profiling and a subgroup of ALCL might arise from innate lymphocyte cells as recently described based on transcriptomic analyses (48, 68). The fact that we observe NPM-ALK–expressing tumor cells that show a surface marker expression pattern that is typical for DN cells, as well as the presence of immature single positive cells suggests, that an immature thymic cell population is targeted for transformation or that a more mature cell population regresses to a progenitor stage during transformation, potentially through direct repression of the T-cell phenotype by the ALK oncogene (46). Again, MYC might be central to this event, in line with data showing that overexpression of MYC, activated AKT and inhibition of intrinsic apoptosis by expression of BCLXL results in rapid transformation of mature CD4 and CD8 SP mouse T cells (96). Expression of reprogrammed DN lymphoma stem cells was recently described for an Lck-dependent NPM-ALK mouse model (97).

The elevated expression of DNMT1 appeared to be an early event after NPM-ALK induction in our model and depletion of DNMT1 completely abrogated lymphomagenesis. Intriguingly, we found DNA methylation signatures, highly resembling human tumors (98), with characteristic CGI hypermethylation and genome-wide hypomethylation as well as DNA hypermethylation of polycomb repressive marks (9). Generally, epigenomic patterns appeared to be highly heterogeneous between individual ALK tumors and showed a loss of cooperative CpG methylation. These observations went hand in hand with up-regulation of DNMTs (DNMT1 and DNMT3b) and TET1, implying that the oncogenic driver NPM-ALK has the potential to interfere with methylation homeostasis and to

induce stochastic methylation aberrations. Interestingly, deletion of DNMT1 did not interfere with cooperative DNA methylation between closely neighboring CpGs because KO and ALKKO cells maintained high correlation between triplet CpGs with a CpG distance less than 20 bp. Thus, we propose that aberrant DNA methylation patterns in tumors might not fully depend on DNMT1 deregulation and that methylation cooperativity at close-by CpGs is independent of the maintenance machinery, but rather dependent on DNMT3a/b and TET enzymes. In general, DNMT3a and DNMT3b primarily bind to methylated CpG-rich regions; however, DNMT3b seems to exhibit additional preferences for actively transcribed genes and correlates with H3K36me3 marks (99). In addition, DNMT3a/b is important to counteract global DNA methylation loss caused by imperfect DNMT1 fidelity during DNA replication in mice (100). De novo methylation occurs more frequently at adjacent CpGs in a distance-dependent manner (99), further indicating that association between neighboring CpGs could be maintained through DNMT3a and DNMT3b in KO and ALKKO samples. Importantly, DNMT3 proteins might be involved in lymphomagenesis through their co-repressor function through interaction with MYC or other transcription factors, even independently of their enzymatic function (101, 102, 103).

In summary, we conclude that ALK-dependent oncogenic pathways result in deregulation of genome-wide DNA methylation patterns during tumorigenesis, which affect important regulatory regions, including lineage-specific transcription factor–binding sites and enhancers of T-cell–specific genes and tumor suppressors. In addition, we suggest that the deregulation of key epigenetic enzymes is a prerequisite to enable tumor formation and loss of maintenance of tumor-specific DNA methylation patterns results in a proliferation block and lack of T-cell transformation.

## Materials and Methods

### Mice

Transgenic mice carrying the human NPM-ALK fusion gene under the T-cell–specific *Cd4* enhancer-promoter system were crossed with mice carrying a conditional T-cell–specific deletion of *Dnmt1* (*Cd4*-Cre–driven recombinase) (31, 53, 104). The NPM-ALK mice were obtained from Lukas Kenner, Department of Pathology, Medical University of Vienna, and the *Dnmt1* knockout mice were obtained from Christian Seiser, Center for Anatomy and Cell Biology, Medical University of Vienna. The genetic background of mice was mixed (C57Bl/6xSV/129). Mice were kept under specific pathogen-free conditions at the Center for Biomedical Research, Medical University of Vienna, and the experiments were carried out in agreement with the ethical guidelines of the Medical University of Vienna and after approval by the Austrian Federal Ministry for Science and Research (BMWF; GZ.: 66.009/0304.WF/V/3b/2014). For genotyping, tissue samples obtained from ear clipping were incubated with tail lysis buffer (100 mM Tris, pH 8.0, 5 mM EDTA, pH 8.0, 200 mM NaCl, and 0.1% SDS) and 40 $\mu$l of proteinase K (10 mg/ml) o/n at 56°C. The next day, 170 $\mu$l of 5M NaCl were added, the solution was centrifuged for 10 min at maximum speed, and the supernatant

was transferred to a new tube. 500 $\mu$l of isopropanol was added and after centrifugation and washing with 70% EtOH, the DNA was dried and dissolved in 150 $\mu$l of sterile water. Genotyping was performed with Promega GoTaq Mastermix according to the manufacturer's suggestions.

### In vivo DNMT inhibitor treatment

NPM-ALK transgenic mice were treated with 1 mg/kg of 5-aza-2′-deoxycytidine (5-aza-CdR) dissolved in PBS administered intraperitoneally two times per week starting at 8 wk of age up to 30 wk of age. After treatment, the mice were monitored three times per week until euthanasia upon signs of sickness and/or tumor development.

### Flow cytometry analysis

Single cell suspensions of thymus, tumor and spleen were obtained by passaging the tissues through a 70-$\mu$m nylon cell strainer in staining buffer (PBS supplemented with 2% FCS and 0.1% sodium azide). $3 \times 10^6$ cells were incubated for 5 min on ice with Fc-block (Pharmingen), after incubation with cell surface markers for 30 min on ice. The cells were washed and fixed with eBioscience FoxP3 Transcription Factor Fixation/Permeabilization Solution for 1 h at 4°C before. Intracellular staining for ALK (D5F3, CST #3633) was performed for 1 h at 4°C followed by incubation with Alexa Fluor 647 goat antimouse IgG antibody (Cat. no. A-21244; Thermo Fisher Scientific) for another hour at 4°C. The cells were washed and resuspended in 100 $\mu$l staining buffer. The cells were measured with a BD Fortessa flow cytometer and analyzed using FlowJo software. The detailed gating strategy is displayed in Fig S8.

### RNA and DNA isolation from murine tumor and thymic tissues

RNA and DNA were isolated using the QIAGEN AllPrep DNA/RNA mini isolation kit to enable simultaneous isolation of nucleic acids from the same specimen. Tissues were homogenized using EPPI-Mikropistills (Schuett-biotec) and RNA and DNA isolation was performed according to the manufacturer's protocol. RNA and DNA were eluted in nuclease free ddH$_2$O.

### Protein extraction and Western blot

For protein extraction from tumors and thymi, the tissue was dounced and homogenized in lysis buffer as previously described (105). Protein concentrations were measured using Bradford and 20 $\mu$g were used for analysis by SDS–PAGE and Western blot as previously described (105). The following antibodies were used for protein expression analysis: ALK (D5F3, CST #3633), pALK (Tyr1278, CST #6941), STAT3 (D3Z2G, CST #12640), pSTAT3 (Tyr705, CST #9145), DNMT1 (H300, sc-20701), CDK4 (C-22, sc-260; Santa Cruz), CDK6 (HPA002637; Sigma-Aldrich), PCNA (ab2426; Abcam), and $\alpha$-TUBULIN (1E4C11, 66031-1-Ig; Proteintech). Goat antirabbit IgG HRP conjugate, JD111036047, and rabbit antimouse IgG HRP conjugated, JD315035008 antibodies were used as secondary antibodies.

## Preparation of nuclear extracts

For isolation of nuclear fractions from tumors and thymi, the tissue was carefully dounced and homogenized in sucrose buffer (0.32 M sucrose, 10 mM Tris–HCl, pH 8.0, 3 mM $CaCl_2$, 2 mM MgOAc, 0.1 mM EDTA, and 0.5% NP-40) containing protease inhibitors using plastic pestles. Nuclei were collected by centrifugation at 500$g$ for 5 min at 4°C and supernatants, containing the cytoplasmic fractions were transferred into new tubes. Nuclei were washed two times with sucrose buffer without NP-40 before they were resuspended in 50 $\mu$l low salt buffer (20 mM Hepes, pH 7.9, 1.5 mM $MgCl_2$, 20 mM KCl, 0.2 mM EDTA, and 25% glycerol [vol/vol]) containing protease inhibitors. An equal amount of high salt buffer (20 mM Hepes, pH 7.9, 1.5 mM $MgCl_2$, 800 mM KCl, 0.2 mM EDTA, and 25% glycerol [vol/vol], 1% NP-40) containing protease inhibitors was added. Samples were incubated for 1 h 30 min at 4°C on a shaker before they were centrifuged at 21,000$g$ at 4°C. Supernatant containing the nuclear fraction was transferred into a new tube and protein concentration was measured using Bradford. 15 $\mu$g of nuclear fractions were used for SDS–PAGE and Western blot analysis as described previously (105). The following antibodies were used for nuclear protein expression analysis: c-MYC (fE5Q6W, CST #18583) and HDAC1 as nuclear loading control.

## Quantitative RT PCR (qRT-PCR)

For qRT-PCR, RNA was isolated as described above and 1 $\mu$g of RNA was used for random hexamer cDNA synthesis using the qScript cDNA Synthesis Kit (Quantabio) according to supplier's protocol. cDNA was diluted to a final concentration of 5 ng/$\mu$l and qRT–PCR was performed with KAPA SYBR FAST qPCR kits (KAPA Biosystems) on a C1000 thermal cycler, CFX96 real-time system (Bio-Rad) using 10 ng of cDNA per reaction. Three biological replicates per genotype were processed. GAPDH was used for normalization. The following primers were used for RNA-sequencing validation:

*Gapdh* fw: 5'-CGACTTCAACAGCAACTCCCACTCTTCC-3'
*Gapdh* rv: 5'-TGGGTGGTCCAGGGTTTCTTACTCCTT-3'
*Cd44* fw: 5'-ATGAAGTTGGCCCTGAGCAA-3'
*Cd44* rv: 5'-GTGTTGGACGTGACGAGGAT-3'
*Dnmt1* fw: 5'-AGGAGAAGCAAGTCGGACAG-3'
*Dnmt1* rv: 5'-CTTGGGTTTCCGTTTAGTGG-3'
*Notch1* fw: 5'-TGGCAGCCTCAATATTCCTT-3'
*Notch1* rv: 5'-CACAAAGAACAGGAGCACGA-3'
*Myc* fw: 5'-AGTGCTGCATGAGGAGACAC-3'
*Myc* rv: 5'-GGTTTGCCTCTTCTCCACAG-3'
*Cdk4* fv: 5'-TCCCAATGTTGTACGGCTGA-3'
*Cdk4* rv: 5'-ACGCATTAGATCCTTAATGGTCTCA-3'
*Cdk6* fw: 5'-CAGCAACCTCTCCTTCGTGA-3'
*Cdk6* rv: 5'-GATCCCTCCTCTTCCCCCTC-3'
*IAP* fw: 5'-ACTAAcTCCTGCTGACTGG-3'
*IAP* rv: 5'-TGTGGCTTGCTCATAGATTAG-3'

## Immunohistochemistry

Tumor and thymic tissues were fixed in 4% paraformaldehyde, dehydrated in ethanol, and embedded in paraffin. 2-$\mu$M sections were cut, attached to slides, dewaxed, and rehydrated. Epitopes were retrieved by heat-treatment in citrate buffer (pH 6.0; DAKO) or Tris–EDTA buffer (pH 9.0; DAKO). Slides were processed and counterstained as previously described (105). Primary antibodies against DNMT1, ALK pSTAT3 (listed above), or Ki67 (14-5698-80; eBioscience) were used for staining. Pictures were taken with a Zeiss Axio10 (Zeiss) microscope and a Gryphax camera (Jenaoptics) and quantification of positive cells was performed using Definiens Tissue Studio 4.2 Software (Definiens Inc.). Stainings were performed in four biological replicates for each genotype. For each biological replicate, four representative pictures were analyzed and counts were averaged. For Ki67 staining, slides were scanned using the Pannoramic 250 Flash III scanner (3DHISTECH) and analyzed using the Definiens software. Non-proliferative areas in thymi were excluded from quantification. Quantification results are shown as means ± SEM. The significance of the differences between mean values was determined by one-way ANOVA followed by pairwise comparisons to the control group using unpaired $t$ tests.

## Immunofluorescence

Tumor and thymic tissues were formalin fixed and paraffin embedded as described above and epitopes were retrieved by heat-treatment in citrate buffer (pH 6.0; DAKO). Slides were washed in 0.1% PBS-Tween 20 (PBS-T), permeabilized in 0.3% Triton X-100 in PBS-T and blocked in blocking solution (10% goat serum in 0.1% Triton X-100 in PBS-T). Primary antibodies against ALK (CST #3363) and Ki67 (14-5698-80; eBioscience) were diluted 1:250 and 1:1,000 in blocking solution and incubated at 4°C overnight. After incubation with secondary antibodies (Alexa Fluor 594 goat antirabbit, Cat. no. A-11012; Invitrogen and Alexa Fluor 488 goat antirat, Cat. no. A-11006; Invitrogen), they were diluted 1:1,000 in blocking solution and the slides were counterstained with DAPI (1:50,000 from 10 mg/ml stock solution; Serva Eletrophoresis) and embedded with geltol (Calbiochem). Two representative pictures of four biological replicates per genotype were taken with LSM 5 Exciter (Zeiss) with the same exposure time for all slides and quantified by blinding pictures and counting of SP and double-positive cells in two 3 × 3 cm squares per picture. Quantification results are shown as means ± SEM. The significance of the differences between mean values was determined by one-way ANOVA followed by pairwise comparisons to the ALK group using unpaired $t$ tests.

## Dot blot analysis of methylated DNA

Genomic DNA was isolated as described above and diluted to a final concentration of 250 ng. DNA was denatured at 100°C for 10 min in 0.4M NaOH and 10 mM EDTA solution and neutralized using 2M ice-cold ammonium acetate, pH 7.0, before it was applied to the dot blot apparatus to spot the DNA on a nitrocellulose membrane, pre-soaked in 6× saline sodium citrate buffer. After washing the membrane in 2× saline sodium citrate buffer, the DNA was UV-crosslinked using (UV Stratalinker 2400; Stratagene). Subsequently, the membrane was blocked for 1 h at room temperature with 5% milk in PBS-T and incubated with the primary antibody directed against 5mC (D3S2Z, CST #28692) overnight at 4°C. After incubation with the secondary antibody (goat anti-rabbit IgG HRP conjugate,

JD111036047), 5mC signal was detected using the ChemiDoc XRS+ Imaging System (Bio-Rad) and analyzed using Image Lab Software (Bio-Rad). Methylene blue staining (0.02% methylene blue in 0.3M sodium acetate) was used for an internal DNA loading control. Membranes were incubated in methylene blue staining solution for 10 min and membranes were destained 3 × 10 min in water. Methylene blue staining was measured and analyzed as described above. Level of 5mC was calculated as a ratio of 5mC to methylene blue signal intensity in three biological replicates per genotype and three technical replicates, respectively.

### RNA-sequencing (RNA-seq)

RNA and DNA were isolated as described above. RNA concentrations were measured on the NanoDrop 2000 (Invitrogen) and 1,000 ng were sent to the Biomedical Sequencing Facility (CeMM). RNA integrity was tested using the Agilent Bioanalyzer. Stranded mRNA-seq (poly-A enrichment) library preparation was performed and sequenced using an Illumina HiSeq3000/4000 platform (50 nucleotide single-end reads).

### RNA-seq data analysis

Reads were quality-controlled using fastQC (106) and pre-processed using trimgalore (http://www.bioinformatics.babraham.ac.uk/projects/trim_galore/) to trim adapter and low quality sequences. The reads were aligned to mouse genome (mm10) and processed further using STAR (107). Differential gene expression levels of the transcripts were quantified by HTSeq (108) and analyzed using the Bioconductor package DESeq2 (109). Genes with an FDR-adjusted $P < 0.05$ and an absolute fold change of two were considered significantly differentially expressed.

To gain insight into the nature of differentially expressed genes in each analysis, the gene set enrichment analysis of significantly deregulated genes between ALK tumors and Ctrl thymi was performed using oncogenic signature gene sets from MSigDB (33).

### Deconvolution strategy

We used the SCDC library (110) based on MuSiC (111) to conduct the bulk RNA-seq data deconvolution with a sc-RNA-seq reference thymic dataset (59) composed of 12 time points (age), 29 cell types, and 36,084 single-nucleus transcriptomes. The method allows deconvoluting a set of bulk RNA-seq samples based on an sc-RNA-seq dataset with the cell type clustering as a reference without pre-selected marker genes. The sc-RNA-Seq reference is used to build a cell-type-specific gene expression signature matrix. This is subsequently used to recover the underlying cell type proportions for the bulk samples using a tree-guided procedure based on a cell type similarity tree (111). A weighted non-negative least square regression framework is used over the previously selected marker genes, weighting each gene by cross-subject and cross cell variation. The general method considers an observed bulk gene expression $\mathbb{Y} \in \mathbb{R}^{N \times M}$ for N genes across M samples, each containing K-cell types. The deconvolution will try to recover two non-negative matrices $\mathbb{B} \in \mathbb{R}^{N \times K}$ (average gene expression levels in each cell type corresponding to the signature matrix) and $\mathbb{P} \in \mathbb{R}^{K \times M}$ (mixing proportions of the K cell type of one sample) such that: $\mathbb{Y} \approx \mathbb{B}\mathbb{P}$. We added some additional routines to the original library such as parallelism capacity, sparse matrix capability, and an improvement in the selection of marker genes.

### Dynamic threshold for markers selection

The non-negative matrix factorization method is sensitive to the selection of marker genes and we, therefore, tested several strategies on simulated data. In the original tool, the selection of the marker genes was performed by applying a Wilcoxon test between each cell type of interest and the remaining single-nucleus transcriptomes in the single-cell data. A global threshold for the adjusted $P$-value is used for all cell types. However, this strategy results in an unbalanced and incomplete set of marker genes, where cell types may have either a very high number, few, or zero marker genes. To solve this issue, we proposed a dynamic threshold for marker selection where we first create multiple bootstrapping samples on all clusters selecting nb clusters as background and run Wilcoxon tests over each sampling and cell type of interest. Finally, we performed an outlier analysis based on the $\log_2(\text{FC})$ value using the dbscan algorithm. For each cluster, we started by selecting genes with an adjusted $P = 0$, if this resulted in no or too few genes we relaxed the threshold to an adjusted $P < 0.05$, and finally, the remaining cases were sorted by the adjusted $P$-value. For each cell cluster we therefore found at least $N_{min}$ and at most $N_{max}$ marker genes. To determine optimal parameters, we extensively tested our selection strategy within the deconvolution procedure on simulated bulk datasets generated from the single cell data with known proportions. We found best deconvolution results when selecting nb = 4 background cell clusters, chosen in nbs = 40 bootstrapping samples, and setting the number of required marker genes per cluster to be between $N_{min} = 28$ and $N_{max} = 35$. With the optimal parameters we obtained Pearson correlation values of >0.99 between the calculated proportions and the real ones and a sum of residuals of 0.0732.

### Deconvolution of bulk data

Our bulk data sample is taken from 18-wk-old mice. We therefore filtered the single-cell dataset to only include data from 4-, 8-, and 24-wk single-cell data. We used a subset of seven cell types corresponding to T-Cell differentiation pseudotime (59): DN (P), DN (Q), DP (P), DP (Q), $\alpha\beta$T (entry), CD8+ T, and CD4+ T. These seven cell types make up 93.4% of the expression found in the complete set of 29 cell clusters in the single-cell data.

The following similarity tree between cell types was used for the tree-guided procedure:

(i) Subcluster 1: DN(P), DN(Q)
(ii) Subcluster 2: DP(P), DP(Q), $\alpha\beta$T(entry)
(iii) Subcluster 3: CD8+ T, CD4+ T

### Differential expression analysis between simulated and real bulk data

We constructed a simulation of bulk data samples based on proportions estimated from the single cell non-tumor samples. We assumed that if there were extra cell types in the ALK tumor replicas these additional cell types should be characterized through

differentially expressed genes between the simulations and the real replicas.

We chose the top 10% most highly expressed genes in the original bulk data plus the 175 marker genes, and performed a differential expression analysis using DESeq2 between the bulk samples and the corresponding replicas in the simulation.

### Source code

For more details, please refer to the R notebook and the source code at: https://github.com/crhisto/thymus_NPM-ALK_notebook.

### RRBS

DNA was isolated as described above. DNA concentration was measured using Qubit dsDNA HS Assay Kit from Invitrogen according to the manufacturer's protocol. In total, 500 ng (25 ng/μl, 20 μl total) of DNA was sent for RRBS analysis (Biomedical Sequencing Facility, CeMM) according to established protocols (112, 113).

### RRBS data analysis

The global changes in methylation, individual CpGs, and clusters of covered CpGs were analyzed using packages from R Bioconductor (114, 115). For CpG level comparison, percentage of methylation of individual CpGs was calculated using the methylKit package (116) and the coverage files from the bismark aligner (117). To prevent PCR bias and increase the power of the statistical tests, we discarded bases with coverage less than 10 in all samples. The bisulfite conversion rate was calculated as the number of thymines (non-methylated cytosines) divided by coverage for each non-CpG cytosine, as implemented in methylKit. Differential methylation analysis of single CpGs between different groups was also performed in the same package and the CpGs with $P < 0.01$ and $\beta$ value differences more than 25% defined as significant. In addition, density, PCA, and correlation plots were also generated by R Bioconductor packages. DMRs were determined using DSS package (118, 119, 120, 121) which outperforms other methods when the sample size per group is small owing to the adoption of the Wald test with shrinkage for determining differentially methylated cytosines (122). We identified DMRs using the coverage files from bismark with a $P$-value threshold of 0.01 and $\Delta\beta$ value more than 25%. The individual CpGs and DMRs were annotated using Annotatr package (123). For the enhancer analyses, DMRs between ALK tumor cells and Ctrl thymocytes were compared with ENCODE datasets for H3K4me1 and H3K27ac marks in murine thymus. We downloaded the ENCODE datasets from the ENCODE portal (75) (www.encodeproject.org) with the following identifiers: ENCFF666XCJ and ENCFF354DWX. Enhancers were predicted, and k-means clustering analysis was performed using deep Tools v2.0 (124). The "computeMatrix" command with the sub-command "scale-regions" was used to generate the table underlying the heat maps, using a 5-kb window indicated by start (S) and end (E) of the DMR region. "plotHeatmap" was used to visualize the table. Region set enrichment analysis against publicly available datasets using the LOLA software was used to identify shared biologically patterns among DMRs (64). Significant CpGs ($P < 0.05$; beta value difference >15%) from

promoter region (hyper-methylated: 850; hypo-methylated: 576) were merged into 200 base pair tiling regions across the genome before LOLA analysis. The significant hyper- and hypomethylated regions obtained were used as the query set and the set of all differentially methylated tilling regions were used as universe set. For a focused analysis, only region sets from ESCs, T lymphocytes, and thymus in the LOLA core database were included. All the enrichments with a FDR adjusted $P < 0.05$ using the Benjamini Hochberg procedure were considered significant. For network analysis, data were analyzed through the use of QIAGEN's Ingenuity Pathway Analysis (IPA, QIAGEN Redwood City, www.qiagen.com/ingenuity).

### Statistics

Data are represented as mean ± SD, if not otherwise stated and were analyzed using GraphPad Prism (version 6, GraphPad Software, Inc.). To assess differences between groups, unpaired $t$ test or one- or two-way ANOVA followed by multiple comparison were used, depending on number of samples. Significance was defined according to following $P$-values: $*P < 0.05$; $**P < 0.01$; $***P < 0.001$ $****P < 0.0001$. Nonsignificant $P$-values are not shown. Survival statistics were analyzed using GraphPad Prism. Pairwise curve comparison with ALK tumors using log-rank calculations (Mantel–Cox test) were used to assess differences between the groups.

## Data Availability

The RNA-seq and RRBS datasets were deposited to the NCBI Gene Expression Omnibus (125) accession number GSE162218. FACS raw data were deposited to http://flowrepository.org (126) ID number FR-FCM-Z35D.

## Supplementary Information

## Acknowledgements

This work was supported by funds of the Austrian Science Foundation (FWF) projects P27616 and I4066 and by the Federal Ministry of Education and Research (BMBF)-funded de.NBI Cloud within the German Network for Bioinformatics Infrastructure (de.NBI) (031A537B, 031A533A, 031A538A, 031A533B, 031A535A, 031A537C, 031A534A, and 031A532B). S Lagger is a fellow of the Postdoc Career Program at Vetmeduni Vienna. The authors thank the International Max Planck Research School for Intelligent Systems for supporting CdJ Cardona. We would like to thank Helga Schachner and Michaela Schlederer for technical support.

### Author Contributions

E Redl: investigation and writing—original draft.
R Sheibani-Tezerji: data curation.
CdJ Cardona: data curation.
P Hamminger: investigation.

G Timelthaler: data curation.
MR Hassler: investigation.
M Zrimsek: investigation.
S Lagger: investigation and writing—review and editing.
T Dillinger: investigation.
L Hofbauer: investigation.
K Draganic: investigation.
A Tiefenbacher: investigation.
M Kothmayer: investigation.
CH Dietz: data curation.
BH Ramsahoye: investigation.
L Kenner: resources.
C Bock: supervision.
C Seiser: resources and supervision.
W Ellmeier: supervision.
G Schweikert: data curation and supervision.
G Egger: conceptualization, data curation, supervision, methodology, project administration, and writing—original draft, review, and editing.

## Conflict of Interest Statement

The authors declare that they have no conflict of interest.

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
