## [Reviewer comments · Life Science Alliance]

Life Science Alliance

Requirement of DNMT1 to orchestrate epigenomic reprogramming for NPM-ALK driven lymphomagenesis

Elisa Redl, Raheleh Sheibani-Tezerji, Christian Cardona, Patricia Hamminger, Gerald Timelthaler, Melanie Hassler, Masa Zrimsek, Sabine Lagger, Thomas Dillinger, Lorena Hofbauer, Kristina Draganic, Andreas Tiefenbacher, Micheal Kothmayer, Charles Dietz, Bernard Ramsahoye, Lukas Kenner, Christoph Bock, Christian Seiser, Wilfried Ellmeier, Gabriele Schweikert, and Gerda Egger
DOI: <https://doi.org/10.26508/lsa.202000794>

Corresponding author(s): Gerda Egger, Medical University of Vienna

Review Timeline:

Submission Date:	2020-05-25
Editorial Decision:	2020-06-26
Revision Received:	2020-11-08
Editorial Decision:	2020-11-24
Revision Received:	2020-11-28
Accepted:	2020-12-01

Scientific Editor: Shachi Bhatt

Transaction Report:

June 26, 2020

Re: Life Science Alliance manuscript #LSA-2020-00794-T

Gerda Egger
Medical University of Vienna
Department of Pathology
Waehringer Guertel 18-20
Vienna 1090
Austria

Dear Dr. Egger,

Thank you for submitting your manuscript entitled "Requirement of DNMT1 to orchestrate epigenomic reprogramming during NPM-ALK driven lymphomagenesis" to Life Science Alliance. The manuscript was assessed by expert reviewers, whose comments are appended to this letter.

As you will see, all referees express an interest in the study and appreciate the comprehensive genome-wide analyses you have performed. Nonetheless, they raise some concerns, which should be addressed in a revised version of the manuscript. Specifically, it will be important to further discuss DNMT1 heterogeneity, its effect on CpG methylation, and how this affects the analyses and conclusions (ref #2- point 1, ref #3- point 3).

In addition, referee #2 notes that the conclusion on the effects of methylation patterns on transcription factor binding is not fully supported by the current data. The referee proposes an additional analysis based on the acquired data. This analysis should either be included, or the text revised to make clear which aspects of the conclusions will require further future experimental testing.

Please also discuss the questions raised by referee #3 regarding ALK and STAT3 phosphorylation (points 1, 2), MYC (ref #1- point 1, ref#3- point 4) and respond to referee #1's point 4, providing additional experimental data where available. In addition, referee #1's question regarding Cyclin D1 expression (point 3) can likely be addressed on the mRNA level using the available RNAseq data and should be responded to. Please also carefully consider all other points raised by the referees and revise the manuscript and add to the discussion accordingly.

Once these issues are resolved we will be happy to consider the study further for publication in Life Science Alliance. Therefore we now invite you to prepare and submit a revised version of your manuscript.

The typical timeframe for revisions is three months. Please note that papers are generally considered through only one revision cycle, so strong support from the referees on the revised version is needed for acceptance. We are aware that many laboratories cannot function at full efficiency during the current COVID-19/SARS-CoV-2 pandemic and have therefore extended our 'scooping protection policy' to cover the period required for a full revision to address the experimental issues highlighted in the editorial decision letter. Please contact the scientific editor handling your manuscript to discuss a revision plan should you need additional time, and also if you see a paper with related content published elsewhere.

Thank you for this interesting contribution to Life Science Alliance. We are looking forward to receiving your revised manuscript.

Sincerely,

Reilly Lorenz
Editorial Office Life Science Alliance
Meyerhofstr. 1
69117 Heidelberg, Germany
t +49 6221 8891 414
e contact@life-science-alliance.org
www.life-science-alliance.org

B. MANUSCRIPT ORGANIZATION AND FORMATTING:

Reviewer #1 (Comments to the Authors (Required)):

The manuscript by E. Redl et al. entitled: "Requirement of DNMT1 to orchestrate epigenomic reprogramming during NPM-ALK driven lymphomagenesis" describes genome-scale changes in mouse thymic T cells induced by NPM-ALK expression driven by CD4 promoter and Cre-induced DNMT1 loss in a transgenic mouse model. NPM-ALK induces change in the thymocytes immunophenotype which occurs only after a period of latency, accompanied by induction of the MYC and NOTCH1 expression and deregulation of key enzymes from the epigenetic machinery. There are aberrant DNA methylation patterns, overlapping with regulatory regions, plus a high degree of epigenetic heterogeneity between individual ALK+ T-cell tumors. In addition, ALK+ tumors show a loss of collaborative methylation patterns of neighboring CpG sites. Strikingly, deletion of DNMT1 completely abrogates lymphomagenesis in this model, despite oncogenic signaling through NPM-ALK, suggesting essential role for this enzyme in NPM-ALK-promoted malignant cell transformation.

This is a very interesting study but some data interpretations require clarification.

Major comments.

1. While the authors propose emergence of NOTCH1-MYC axis, the evidence is so far merely correlative. MYC expression and activation is promoted by various pathways; NPM-ALK could be one of them, in particular when high expression of the kinase is achieved in the thymic cells. Can nuclear MYC protein expression be evaluated in the pre-tumor and tumor stage in ALK+ thymocytes?
2. RNA-Seq data of the ALK+ thymocytes pre-tumor stage would have been informative
3. Because CDK4 and CDK6 mRNA are upregulated, one wonders if the same is true for Cyclin D1- on both mRNA and protein levels, identified as aberrantly expressed in human ALK+ tumors.
4. While the authors postulate ALK-mediated "dedifferentiation" to DN stage, thymocyte-stage independent inhibition by NPM-ALK per se of expression of DP phenotype in the CD4-expressing thymocytes should be considered as an important alternative, given the phenotype of human ALK+ lymphomas.
5. Fig. 5: having results to compare on pre-tumor ALK+ thymocytes would have been helpful.

Minor comments.

1. Perhaps revised title such as: "Requirement for DNMT1 to orchestrate epigenomic reprogramming in the NPM-ALK driven lymphomagenesis" would convey better the key message of the manuscript.
2. While the term collaborative methylation coined by others is used, the term associated methylation may be more appropriate, given the uncertain molecular underpinning of the phenomenon.
3. The phrase on page 7 regarding ALK induced transformation mechanisms affecting cell tumorigenic pathways seems misplaced and fitting better the previous section on tumorigenic pathways.
4. The phrase on page 8 regarding mTOR-regulated sounds a bit awkward. Perhaps, stating that these are mTORC1-regulated genes would be better.

Reviewer #2 (Comments to the Authors (Required)):

In this manuscript Redl et al used a T cell-specific lymphoma model to interrogate the transcriptional and epigenetic changes that lead to transformation. First they perform DGE using RNA seq samples from ALK tumor cells and thymocytes from wt mice and identify the MYC pathway as one of the top unregulated pathways in ALK. They further identify DNMT1 to be upregulated in the ALK tumors. They convincingly show that loss of DNMT1 abrogates lymphomagenesis by using conditional deletion of DNMT1, suggesting that DNMT1 is essential for tumorigenesis. They furthermore report aberrant DNA methylation and increased heterogeneity of methylation at regulatory regions.

The results obtained from the conditional DNMT1 deletion are very convincing, supporting a role for this enzyme in tumorigenesis.

However, I have one major concern related to the genomic analysis related to DNA methylation: Figure 3D shows staining for DNMT1 in controls and KOs and tumors, etc... The staining for DNMT1 in the control sample is heterogenous, suggesting that some cells have very high DNMT1 levels and others lack DNMT1. I am not sure how to interpret this, but this reflects the cell heterogeneity and changes in cell composition in ALK - as also observed using surface markers in Fig2. I wonder how much this heterogeneity and the changes in composition influence the results obtained from the genomics experiments that use the bulk population to calculate differences in DNA Methylation? Could the authors comment on this or perform experiments that address how cell heterogeneity, more specifically differences in cell composition between the analysed samples, could influence the results?

The authors summarise in their discussion that "we conclude that ALK dependent oncogenic pathways result in deregulation of genome-wide DNA methylation patterns during tumorigenesis, that affect important regulatory regions including lineage specific transcription factor binding sites ...". That suggests that indeed DNA methylation would block such transcription factors. I am wondering, if based on the analysis they have already performed on the methylation data, they could actually look at such TF binding sites and test if methylation is indeed changed, or include motifs enrichments at the called DMRs to support such statements?

Reviewer #3 (Comments to the Authors (Required)):

In this manuscript, Redl, et al present very interesting data from a genetically modified mouse model of NPM-ALK driven T-cell Lymphoma demonstrating latency of the oncogenic effect of NPM-ALK during T-cell maturation and significant aberrations in DNA methylation accompanying tumor formation.

The authors also show striking data in which knockout of DNMT-1 abrogates the tumorigenic effect of the NPM-ALK fusion. Not unexpectedly the NOTCH1-MYC pathway is implicated in the effects of NPM-ALK on oncogenesis. Further analysis of the DNA methylation patterns showed heterogeneity among tumors and no effect of DNMT1 deletion on close by CpG methylation patterns.

Overall the experiments are well designed and the data are of high quality.

There are some aspects of the results and discussion which if addressed would add to the overall impact of the paper.

1. On p.9 the authors state that with tumor progression the levels of ALK and P-ALK increase, while the western blot in Fig 2B shows considerable overlap in detected ALK levels in small versus end stage tumors.

2. In Figure 3D the pSTAT level appears to go down in the DNMT1 knockout cells(although not so clearly in the western blot in Fig 3E). If this effect is real, could it be a somewhat independent effect of DNMT1 on STAT3 phosphorylation?

3.. Given the mixed results of the effects of DNMT1 on CpG methylation patterns and the overall heterogeneity it is possible that DNMT3 may have effects other than through its enzymatic methylation function (eg as in comment 2, or as a co-repressor, as has been suggested by others). This point deserves some mention in the discussion.

4. While perhaps outside the scope of this manuscript, it would be interesting to see data showing that reduction of MYC expression by knockdown would slow or prevent T-cell lymphoma tumorigenesis induced by NPM-ALK

Likewise, given the dramatic effect of DNMT1 knockout on NPM-ALK induced lymphoma formation, it would be interesting to see if decitabine treatment in this animal model would at least attenuate tumor formation.

If so, it would have important clinical implications.

We are very thankful for the constructive comments of the reviewers and wish to address their suggestions in the following point-by-point reply. We have highlighted specific remarks in bold that were also emphasized by the editor. The revised manuscript includes additional experimental and bioinformatics analyses as suggested by the reviewers. We think that the critical suggestions have greatly improved our manuscript and we hope that the reviewers now find it acceptable for publication in Life Science Alliance.

Reviewer #1:

The manuscript by E. Redl et al. entitled: "Requirement of DNMT1 to orchestrate epigenomic reprogramming during NPM-ALK driven lymphomagenesis" describes genome-scale changes in mouse thymic T cells induced by NPM-ALK expression driven by CD4 promoter and Cre-induced DNMT1 loss in a transgenic mouse model. NPM-ALK induces change in the thymocytes immunophenotype which occurs only after a period of latency, accompanied by induction of the MYC and NOTCH1 expression and deregulation of key enzymes from the epigenetic machinery. There are aberrant DNA methylation patterns, overlapping with regulatory regions, plus a high degree of epigenetic heterogeneity between individual ALK+ T-cell tumors. In addition, ALK+ tumors show a loss of collaborative methylation patterns of neighboring CpG sites. Strikingly, deletion of DNMT1 completely abrogates lymphomagenesis in this model, despite oncogenic signaling through NPM-ALK, suggesting essential role for this enzyme in NPM-ALK-promoted malignant cell transformation.

This is a very interesting study but some data interpretations require clarification.

Major comments.

1. While the authors propose emergence of NOTCH1-MYC axis, the evidence is so far merely correlative. MYC expression and activation is promoted by various pathways; NPM-ALK could be one of them, in particular when high expression of the kinase is achieved in the thymic cells. Can nuclear MYC protein expression be evaluated in the pre-tumor and tumor stage in ALK+ thymocytes?

Several studies have implicated Myc as a downstream target of NPM-ALK in ALCL (PMID: 29609590; PMID: 28659618; PMID: 12213716) or the related MYCN gene in ALK dependent neuroblastoma (PMID: 22439933). As requested by the reviewer, we have performed a Western blot on nuclear extracts of Ctrl, NPM-ALK tumor free as well as early stage tumors and end stage tumors, demonstrating a massive upregulation of MYC expression in early stage tumors. These data have been added to the revised manuscript in Fig. 2H

2. RNA-Seq data of the ALK+ thymocytes pre-tumor stage would have been informative

We agree with the reviewer that RNA seq analyses might have added important insights. Unfortunately, we have currently not the possibility to perform these additional experiments.

3. Because CDK4 and CDK6 mRNA are upregulated, one wonders if the same is

true for Cyclin D1- on both mRNA and protein levels, identified as aberrantly expressed in human ALK+ tumors.

We analyzed Cyclin D1 expression on protein and RNA level using western blots and RNA-seq data. Both protein and RNA levels appear generally elevated in ALK positive cells, although some heterogeneity in expression levels were obvious at different stages of ALK tumorigenesis. The western blot for Cyclin D1 was added in Fig.2I, the RNA-seq expression in Suppl.Fig.6D.

4. While the authors postulate ALK-mediated "dedifferentiation" to DN stage, thymocyte-stage independent inhibition by NPM-ALK per se of expression of DP phenotype in the CD4-expressing thymocytes should be considered as an important alternative, given the phenotype of human ALK+ lymphomas.

We added the following sentence referencing the study by Ambrogio et al PMID: 19887607 and a recent study by Congras et al PMID: 33141118 (page 8):

"Together, these data suggest that ALK tumors are developing from a subpopulation of immature T cells or through reprogramming of DP T cells towards the DN stage based on the direct repression of the T cell phenotype by NPM-ALK as previously suggested (46,47)."

5. Fig. 5: having results to compare on pre-tumor ALK+ thymocytes would have been helpful.

As mentioned above, we agree but regret that we have no possibility to add those data.

Minor comments.

1. Perhaps revised title such as: "Requirement for DNMT1 to orchestrate epigenomic reprogramming in the NPM-ALK driven lymphomagenesis" would convey better the key message of the manuscript.

We changed the title to: "Requirement of DNMT1 to orchestrate epigenomic reprogramming for NPM-ALK driven lymphomagenesis".

2. While the term collaborative methylation coined by others is used, the term associated methylation may be more appropriate, given the uncertain molecular underpinning of the phenomenon.

We changed the term "collaborative" to "associated".

3. The phrase on page 7 regarding ALK induced transformation mechanisms affecting cell tumorigenic pathways seems misplaced and fitting better the previous section on tumorigenic pathways.

We shifted the sentence to the previous section as suggested.

4. The phrase on page 8 regarding mTOR-regulated sounds a bit awkward. Perhaps, stating that these are mTORC1-regulated genes would be better.

We rephrased the sentence to "mTORC1-regulated genes".

Reviewer #2

In this manuscript Redl et al used a T cell-specific lymphoma model to interrogate the transcriptional and epigenetic changes that lead to transformation. First they perform DGE using RNA seq samples from ALK tumor cells and thymocytes from wt mice and identify the MYC pathway as one of the top unregulated pathways in ALK. They further identify DNMT1 to be upregulated in the ALK tumors. They convincingly show that loss of DNMT1 abrogates lymphomagenesis by using conditional deletion of DNMT1, suggesting that DNMT1 is essential for tumorigenesis. They furthermore report aberrant DNA methylation and increased heterogeneity of methylation at regulatory regions.

The results obtained from the conditional DNMT1 deletion are very convincing, supporting a role for this enzyme in tumorigenesis.

However, I have one major concern related to the genomic analysis related to DNA methylation:

Figure 3D shows staining for DNMT1 in controls and KOs and tumors, etc... The staining for DNMT1 in the control sample is heterogenous, suggesting that some cells have very high DNMT1 levels and others lack DNMT1. I am not sure how to interpret this, but this reflects the cell heterogeneity and changes in cell composition in ALK - as also observed using surface markers in Fig2. I wonder how much this heterogeneity and the changes in composition influence the results obtained from the genomics experiments that use the bulk population to calculate differences in DNA Methylation? Could the authors comment on this or perform experiments that address how cell heterogeneity, more specifically differences in cell composition between the analysed samples, could influence the results?

The heterogeneity of DNMT1 expression is a consequence of different proliferation states of thymocytes in the control thymus. DNMT1 is a cell cycle regulated gene and thus highly expressed in transformed cells and proliferating thymocytes, yet silenced in non-proliferating thymocyte subsets. Rodriguez et al showed that thymocyte differentiation in human involves demethylation of individual lineage specific transcription factors (mainly in CpG poor regions) but does not involve large scale epigenomic reprogramming (PMID: 25539926).

In order to address the potential cellular heterogeneity, we used a deconvolution strategy to dissect the differences in cell composition in the different genotypes based on recently published single cell RNA-seq data (PMID: 32079746). These analyses demonstrate that Ctrl, KO and ALKKO samples show similar cell compositions, whereas ALK tumor samples show enriched DN cell populations and resemble more embryonic stage thymi, which is in line with our FACS data. The deconvolution analysis was added as Fig. 6 in the revised manuscript. Therefore, we assume that the analysis of bulk samples in our study can be used to provide insights into tumor specific transcriptional and epigenomic pathways.

The authors summarise in their discussion that "we conclude that ALK dependent oncogenic pathways result in deregulation of genome-wide DNA methylation patterns during tumorigenesis, that affect important regulatory regions including lineage specific transcription factor binding sites ... ". That suggests that indeed DNA methylation would block such transcription factors. I am wondering, if based on the analysis they have already performed on the methylation data, they could actually look at such TF binding sites and test if methylation is

indeed changed, or include motifs enrichments at the called DMRs to support such statements?

In order to address this question, we re-ran the LOLA analysis, focusing on 200 bp tiling windows in promoter associated hyper- and hypomethylated DMRs. This analysis again shows high association of hypermethylated regions with Polycomb repressive complexes. Hypomethylated regions overlap significantly with transcription factor enriched regions including JUN, STAT3, BATF and other T cell specific TFs. We used the same 200bp tiling regions and performed motif enrichment analysis with the MEME Suite tool AME (analysis of motif enrichment) using the TF motifs associated with the TFs identified by LOLA. We find significant enrichment of STAT3 and some T cell specific factors both in the hypo and hypermethylated regions, suggesting that those regions are prone to changes in DNA methylation, most likely due to altered regulation of gene expression in tumor cells. The new LOLA and AME analyses can be found in Fig. 7H and Suppl.table 5.

Reviewer #3

In this manuscript, Redl, et al present very interesting data from a genetically modified mouse model of NPM-ALK driven T-cell Lymphoma demonstrating latency of the oncogenic effect of NPM-ALK during T-cell maturation and significant aberrations in DNA methylation accompanying tumor formation. The authors also show striking data in which knockout of DNMT-1 abrogates the tumorigenic effect of the NPM-ALK fusion. Not unexpectedly the NOTCH1-MYC pathway is implicated in the effects of NPM-ALK on oncogenesis. Further analysis of the DNA methylation patterns showed heterogeneity among tumors and no effect of DNMT1 deletion on close by CpG methylation patterns. Overall the experiments are well designed and the data are of high quality. There are some aspects of the results and discussion which if addressed would add to the overall impact of the paper.

1. On p.9 the authors state that with tumor progression the levels of ALK and P-ALK increase, while the western blot in Fig 2B shows considerable overlap in detected ALK levels in small versus end stage tumors.

We apologize for the confusion. We meant to say "tumor development". Indeed, early stage tumors show already highly elevated pALK levels. We corrected this accordingly in the manuscript.

2. In Figure 3D the pSTAT level appears to go down in the DNMT1 knockout cells(although not so clearly in the western blot in Fig 3E). If this effect is real, could it be a somewhat independent effect of DNMT1 on STAT3 phosphorylation?

We re-analyzed the immunohistology stainings to assess this question. However, it appears that there are higher pSTAT3 levels in the medullary areas of the ALKKO thymi as compared to the cortex. Those areas are shown also in the figure and correspond well with ALK tumor levels. Thus, we think that overall pSTAT3 levels, as also seen in the western blot analyses, are comparable.

3. Given the mixed results of the effects of DNMT1 on CpG methylation patterns and the overall heterogeneity it is possible that DNMT3 may have effects other than through its enzymatic methylation function (eg as in comment 2, or as a co-repressor, as has been suggested by others). This point deserves some mention in the discussion.

We added the following sentence to the discussion section referencing PMID: 15616584, PMID: 11350943, PMID: 19786833:

"Importantly, DNMT3 proteins might be involved in lymphomagenesis through their co-repressor function through interaction with MYC or other transcription factors, even independently of their enzymatic function (101-103)."

4. While perhaps outside the scope of this manuscript, it would be interesting to see data showing that reduction of MYC expression by knockdown would slow or prevent T-cell lymphoma tumorigenesis induced by NPM-ALK Likewise, given the dramatic effect of DNMT1 knockout on NPM-ALK induced lymphoma formation, it would be interesting to see if decitabine treatment in this animal model would at least attenuate tumor formation. If so, it would have important clinical implications.

Regarding MYC, we are currently constructing CRISPR/Cas9 vectors, to follow up on MYCs effects in more detail in a subsequent study. Regarding DNMT1 inhibitor treatment, we added Suppl. Figure 5, showing that decitabine treatment can indeed

attenuate tumor formation.

November 24, 2020

RE: Life Science Alliance Manuscript #LSA-2020-00794-TR

Prof. Gerda Egger
Medical University of Vienna
Department of Pathology
Waehringer Guertel 18-20
Vienna 1090
Austria

Dear Dr. Egger,

Thank you for submitting your revised manuscript entitled "Requirement of DNMT1 to orchestrate epigenomic reprogramming for NPM-ALK driven lymphomagenesis". We would be happy to publish your paper in Life Science Alliance pending final revisions necessary to meet our formatting guidelines.

Along with the points listed below, please also attend to the following,

- please use the [10 author names, et al.] format in your references (i.e. limit the author names to the first 10)
- please add scale bars to Figure 2B, 4B

A. FINAL FILES:

-- Summary blurb (enter in submission system): A short text summarizing in a single sentence the study (max. 200 characters including spaces). This text is used in conjunction with the titles of papers, hence should be informative and complementary to the title. It should describe the context

and significance of the findings for a general readership; it should be written in the present tense and refer to the work in the third person. Author names should not be mentioned.

B. MANUSCRIPT ORGANIZATION AND FORMATTING:

Sincerely,

Shachi Bhatt, Ph.D.
Executive Editor
Life Science Alliance
<https://www.lsjournal.org/>
Tweet @SciBhatt @LSAJournal

Reviewer #1 (Comments to the Authors (Required)):

No additional comments.

Reviewer #2 (Comments to the Authors (Required)):

The authors have addressed all my concerns.

December 1, 2020

RE: Life Science Alliance Manuscript #LSA-2020-00794-TRR

Prof. Gerda Egger
Medical University of Vienna
Department of Pathology
Waehringer Guertel 18-20
Vienna 1090
Austria

Dear Dr. Egger,

Thank you for submitting your Research Article entitled "Requirement of DNMT1 to orchestrate epigenomic reprogramming for NPM-ALK driven lymphomagenesis". It is a pleasure to let you know that your manuscript is now accepted for publication in Life Science Alliance. Congratulations on this interesting work.

DISTRIBUTION OF MATERIALS:

Again, congratulations on a very nice paper. I hope you found the review process to be constructive and are pleased with how the manuscript was handled editorially. We look forward to future exciting submissions from your lab.

Sincerely,

Shachi Bhatt, Ph.D.

Executive Editor

Life Science Alliance

<https://www.lsjournal.org/>
